# FLoRG: Federated Fine-tuning with Low-rank Gram Matrices and Procrustes Alignment

**Chuiyang Meng** [*]
The University of British Columbia
`chuiyangmeng@ece.ubc.ca`

**Ming Tang**
Southern University of Science and Technology
`tangm3@sustech.edu.cn`

**Vincent W.S. Wong**
The University of British Columbia
`vincentw@ece.ubc.ca`

## Abstract

Parameter-efficient fine-tuning techniques such as low-rank adaptation (LoRA) enable large language models (LLMs) to adapt to downstream tasks efficiently. Federated learning (FL) further facilitates this process by enabling collaborative fine-tuning across distributed clients without sharing private data. However, the use of two separate low-rank matrices in LoRA for federated fine-tuning introduces two types of challenges. First, aggregation error can arise from separately aggregating the two low-rank matrices. Second, even if the server aggregates the product of two low-rank matrices, it needs to decompose the aggregated matrix back into low-rank matrices. Since the decomposition is not unique, it can lead to decomposition drift. To tackle the aforementioned challenges, we propose federated low-rank Gram-matrix aggregation (FLoRG), a federated fine-tuning framework which employs a single low-rank matrix for fine-tuning and aggregates its Gram matrix (i.e., the matrix of inner products of its column vectors). FLoRG can eliminate the aggregation error and reduce the communication overhead. It also minimizes the decomposition drift by introducing a Procrustes alignment approach which aligns the decomposed matrix between consecutive fine-tuning rounds for consistent updates. We theoretically analyze the convergence of FLoRG and prove that adopting the Procrustes alignment results in a tighter convergence bound. Experimental results across multiple LLM fine-tuning benchmarks demonstrate that FLoRG outperforms five state-of-the-art baseline schemes by providing higher downstream task accuracy and can reduce the communication overhead by up to $2041\times$.

## 1 Introduction

Large language models (LLMs) (Zhao et al., 2023; Achiam et al., 2023; Touvron et al., 2023; Grattafiori et al., 2024) have achieved state-of-the-art performance across a wide range of natural language processing tasks. However, their massive scale introduces critical challenges in terms of computation cost, memory consumption, and adaptability to downstream tasks. To address these concerns, low-rank adaptation (LoRA) (Hu et al., 2022; Hayou et al., 2024; Kopiczko et al., 2024; Liu et al., 2024) has emerged as an effective approach. In particular, the LoRA module employs a fine-tuning matrix $\Delta \mathbf{W}$ with two low-rank matrices $\mathbf{B}$ and $\mathbf{A}$ into the pretrained model $\mathbf{W}^0$ as $\mathbf{W} = \mathbf{W}^0 + \Delta \mathbf{W} = \mathbf{W}^0 + \mathbf{BA}$. Thus, it enables task-specific adaptation while only updating matrix $\Delta \mathbf{W}$. This approach reduces both memory usage and computation overhead significantly when compared with full-model fine-tuning. However, fine-tuning still favors domain-specific data at scale. Such data is typically distributed across multiple clients and thus requires collaborative fine-tuning. To resolve this issue, federated learning (FL) (McMahan et al., 2017; Li et al., 2020) provides a privacy-preserving framework for collaborative model training, where multiple clients fine-tune a shared global model without exposing their raw data. Combining LoRA with FL is therefore highly

---

[*]Corresponding author: Chuiyang Meng

appealing: clients can collaboratively fine-tune a model by locally performing lightweight training via LoRA modules and uploading the low-rank matrix updates for global aggregation.

The conventional works (Zhang et al., 2024; Fang et al., 2024; Zhang et al., 2023b; Wu et al., 2024) propose federated fine-tuning with LoRA, which enables each client $n$ to transmit its low-rank matrices $\mathbf{B}_n$ and $\mathbf{A}_n$ to the central server. Afterwards, the central server aggregates $\mathbf{B}_n$ and $\mathbf{A}_n$ separately and then broadcasts the two aggregated matrices back to each client for performing fine-tuning in the subsequent rounds. In this case, the updated LoRA module after model aggregation can be expressed as $(\frac{1}{N}\sum_{n=1}^{N}\mathbf{B}_n)(\frac{1}{N}\sum_{n=1}^{N}\mathbf{A}_n)$, where $N$ is the number of clients. This approach introduces a challenge: *The aggregation is fundamentally biased,* because the true update should be $\frac{1}{N}\sum_{n=1}^{N}(\mathbf{B}_n\mathbf{A}_n)$, which is different from $(\frac{1}{N}\sum_{n=1}^{N}\mathbf{B}_n)(\frac{1}{N}\sum_{n=1}^{N}\mathbf{A}_n)$. This mismatch introduces a systematic aggregation error which affects the global model update in each fine-tuning round. As the number of rounds increases, the aggregation error is accumulated, which can degrade the fine-tuning performance.

To alleviate the error induced by model aggregation, some works (Yan et al., 2024; Bai et al., 2024; Yan et al., 2025) calculate the product of $\mathbf{B}_n\mathbf{A}_n$ and perform aggregation at the central server. Then, the central server performs matrix decomposition to recover two low-rank matrices for the next fine-tuning round. Although this approach can eliminate the error induced by separately aggregating matrices $\mathbf{B}_n$ and $\mathbf{A}_n$, it introduces another non-trivial challenge: *decomposition is generally not unique.* In particular, the rank of matrices $\mathbf{B}_n$ and $\mathbf{A}_n$ module is typically much smaller than the dimension of the input or output of the parameter matrix. The aggregated matrix is typically rank-deficient and may also exhibit repeated eigenvalues. As a result, there may be multiple valid decompositions into two low-rank matrices. In addition, when the aggregated matrix has eigenvalue multiplicities, many valid decompositions exist. As a result, choosing different matrix decompositions fundamentally changes two low-rank matrices. It incurs a drift in the parameter subspace and changes the direction of the model update for the subsequent fine-tuning rounds. This drift will accumulate as the fine-tuning proceeds and degrade the fine-tuning performance. Furthermore, direct decomposition (e.g., eigendecomposition) may incur rank mismatch since the rank of the aggregated matrix may be different from the local low-rank matrices.

Based on the aforementioned discussions, we focus on addressing the following question: *Is there a federated fine-tuning approach which can eliminate the error induced by separate model aggregation while minimizing the drift induced by matrix decomposition?*

To address this question, we start by rethinking what to aggregate in federated fine-tuning. As LoRA involves matrix multiplication, either separate model aggregation or matrix decomposition is unavoidable. Therefore, one of our key insights is to reparameterize the LoRA module with a single low-rank Gram matrix. We aim to propose a federated fine-tuning framework which aggregates the corresponding Gram matrices to achieve an unbiased aggregation with a low communication overhead. Furthermore, another insight is to propose a Procrustes alignment approach to the decomposed matrix in order to stabilize the fine-tuning while preserving its Gram matrix, thereby mitigating the decomposition drift due to the non-uniqueness of matrix decomposition.

Designing such a framework is challenging due to the following unexplored questions: (i) *How can we design a low-rank parameterization which adapts to any pretrained matrix while supporting error-free aggregation?* (ii) *How to optimize the Procrustes alignment matrix to minimize the decomposition drift?* (iii) *How to characterize the overall convergence rate of the proposed framework under nonconvex losses, and disentangle the impact of Procrustes alignment on the convergence?* In this work, we make the following contributions to address the aforementioned questions:

- We propose FLoRG, a federated fine-tuning framework with a single low-rank matrix and Gram-matrix aggregation. By leveraging a shared semi-orthogonal basis, FLoRG adapts to parameter matrices with arbitrary dimensions. Each client only updates the single low-rank matrix, and the server aggregates the corresponding Gram matrix. FLoRG eliminates the aggregation error by turning the bilinear server-side aggregation into a linear operation. By transmitting a single matrix instead of two matrices, FLoRG is communication-efficient when compared with some of the existing federated LoRA schemes.

- We propose a Procrustes alignment approach after matrix decomposition to preserve the aggregated Gram matrix while aligning the decomposed matrix across fine-tuning rounds to mitigate the decomposition drift. We formulate an optimization problem to minimize

the decomposition drift via a Frobenius-norm objective. The closed-form optimal solution projects the decomposed matrix onto a target $r$-rank subspace without changing its Gram matrix.

- We theoretically analyze the convergence rate of our proposed FLoRG in the nonconvex loss setting. In particular, incorporating our proposed Procrustes alignment mitigates the decomposition drift, thereby resulting in a tighter convergence bound.

- We conduct extensive experiments on GLUE (Wang et al., 2018) with MRPC, QQP, MNLI, QNLI, WNLI, and RTE datasets. We compare our proposed FLoRG with five state-of-the-art baseline schemes, including FedIT (Zhang et al., 2024), FeDeRA (Yan et al., 2024), FFA-LoRA (Sun et al., 2024), FedSA-LoRA (Guo et al., 2025), and FedEx-LoRA (Singhal et al., 2025). Results show that our proposed FLoRG achieves a higher testing accuracy than the baseline schemes under different settings and reduces the communication overhead by up to $2041\times$.

## 2 RELATED WORKS

**LoRA:** Low-rank adaptation (LoRA) (Hu et al., 2022) introduces two low-rank matrices to the pretrained model to perform parameter-efficient fine-tuning. Multiple LoRA variants have been proposed (Zhang et al., 2023a; Dettmers et al., 2023; Hayou et al., 2024; Kopiczko et al., 2024; Liu et al., 2024; Zhao et al., 2024; Bensaïd et al., 2025). For example, in (Zhang et al., 2023a), the authors proposed a LoRA framework to adaptively allocate the parameter budget among weight matrices based on the importance score. In (Zhao et al., 2024), the authors projected the gradient matrix into a low-rank form to perform efficient fine-tuning. In (Bensaïd et al., 2025), the authors reformulated the low-rank adaptation with a single matrix. While the aforementioned works have shown an improvement in the performance, they are primarily designed for centralized settings. Domain-specific data are often possessed by a number of distributed clients, which motivates the incorporation of FL.

**Federated Fine-tuning with LoRA:** LoRA has been incorporated into FL to enable collaborative fine-tuning across distributed clients. The conventional federated fine-tuning works (Zhang et al., 2024; Fang et al., 2024; Zhang et al., 2023b; Wu et al., 2024; Long et al., 2024; Cho et al., 2024; Byun & Lee, 2025) directly aggregate two low-rank matrices separately to obtain the global model. Some works (Babakniya et al., 2023; Yan et al., 2024; Bai et al., 2024; Yan et al., 2025) aggregate the LoRA modules (i.e., the products of two matrices) and then perform matrix decomposition to recover two low-rank matrices. In addition, the authors in (Wang et al., 2024) proposed a stacking-based approach to aggregate the low-rank matrices. The authors in (Sun et al., 2024) proposed to freeze matrix $\mathbf{A}$ and only update matrix $\mathbf{B}$. The authors in (Guo et al., 2025) proposed to let the clients locally update matrix $\mathbf{B}$ and only share matrix $\mathbf{A}$ for aggregation.

## 3 METHODOLOGY

### 3.1 FLoRG

In this section, we propose a federated fine-tuning framework called FLoRG. FLoRG employs a single low-rank matrix and aggregates Gram matrices to eliminate the error induced by separate aggregation as in conventional LoRA. FLoRG performs Procrustes alignment to the decomposed matrix to minimize the decomposition drift. A schematic illustration is shown in Fig. 1.

We consider a central server and $N$ clients. Let $\mathcal{T} = \{1, 2, \ldots, T\}$ and $\mathcal{N} = \{1, 2, \ldots, N\}$ denote the set of $T$ fine-tuning rounds and the set of $N$ clients, respectively. At the beginning of the first fine-tuning round, each client $n \in \mathcal{N}$ has the same pretrained weight matrix $\mathbf{W}^0 \in \mathbb{R}^{d_{\text{out}} \times d_{\text{in}}}$. Note that $\mathbf{W}^0$ is kept frozen and will not be updated during fine-tuning. We consider that each client $n$ has a local dataset $\mathcal{D}_n$. Let $\boldsymbol{\xi}_n \sim \mathcal{D}_n$ denote a mini-batch of training samples. Let $F_n(\mathbf{W}^t; \boldsymbol{\xi}_n)$ denote the local loss function of client $n$ on $\boldsymbol{\xi}_n$ with model $\mathbf{W}^t$ in the $t$-th fine-tuning round. We denote the expected loss of client $n$ with model $\mathbf{W}^t$ as $f_n(\mathbf{W}^t) = \mathbb{E}_{\boldsymbol{\xi}_n \sim \mathcal{D}_n}[F_n(\mathbf{W}^t; \boldsymbol{\xi}_n)]$. Let $\nabla F_n(\mathbf{W}^t; \boldsymbol{\xi}_n) \in \mathbb{R}^{d_{\text{out}} \times d_{\text{in}}}$ denote the local stochastic gradient of client $n$ with model $\mathbf{W}^t$ in the $t$-th fine-tuning round. The learning procedure of FLoRG can be summarized in the following steps.

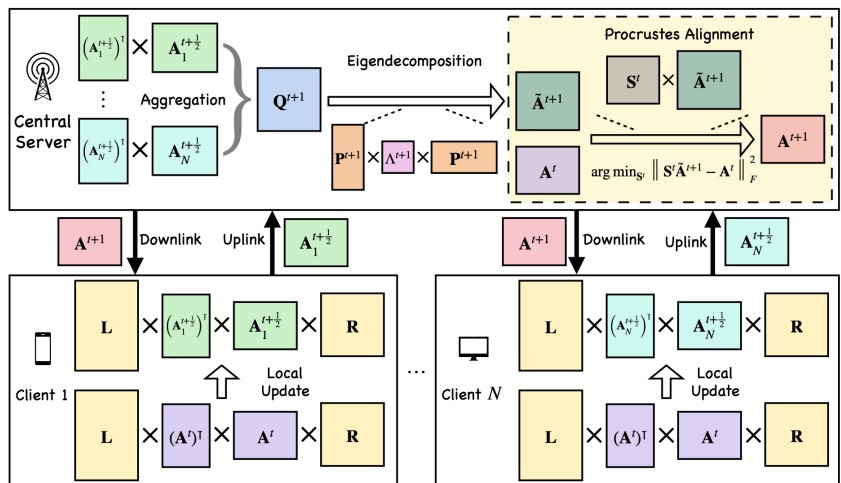

Figure 1: System model of our proposed FLoRG.

**(i) Local Update with Low-rank Gram Matrices:** Different from LoRA, FLoRG uses a single matrix $\mathbf{A}^t \in \mathbb{R}^{r \times k}$ for fine-tuning, where $k = \min\{d_{\text{in}}, d_{\text{out}}\}$ and $r \ll \min\{d_{\text{in}}, d_{\text{out}}\}$ denotes the rank of $\mathbf{A}^t$. Matrices $\mathbf{L} \in \mathbb{R}^{d_{\text{out}} \times k}$ and $\mathbf{R} \in \mathbb{R}^{k \times d_{\text{in}}}$ are initialized and shared across all clients. Both matrices $\mathbf{L}$ and $\mathbf{R}$ are semi-orthogonal, i.e., $\mathbf{L}^\intercal \mathbf{L} = \mathbf{I}_k$ and $\mathbf{R}\mathbf{R}^\intercal = \mathbf{I}_k$, where $\mathbf{I}_k$ denotes the $k \times k$ identity matrix. Note that matrices $\mathbf{L}$ and $\mathbf{R}$ remain unchanged during fine-tuning. The fine-tuning matrix in the $t$-th fine-tuning round is

$$\Delta \mathbf{W}^t = \mathbf{L}\mathbf{Q}^t\mathbf{R} = \mathbf{L}(\mathbf{A}^t)^\intercal \mathbf{A}^t \mathbf{R}, \quad t \in \mathcal{T}, \tag{1}$$

where $\mathbf{Q}^t$ denotes the square Gram parameter matrix in the $t$-th fine-tuning round. By leveraging $\mathbf{L}$ and $\mathbf{R}$, FLoRG is compatible with parameter matrices of any dimensions. Thus, the full model can be expressed as $\mathbf{W}^t = \mathbf{W}^0 + \Delta \mathbf{W}^t$.

In the $t$-th fine-tuning round, the central server broadcasts $\mathbf{A}^t$ to all clients. Client $n \in \mathcal{N}$ updates $\mathbf{A}^t$ using its local dataset. Let $\nabla_{\mathbf{A}} F_n(\mathbf{W}^t; \boldsymbol{\xi}_n) \in \mathbb{R}^{r \times k}$ denote the gradient of the low-rank matrix, which is given by

$$\nabla_{\mathbf{A}} F_n(\mathbf{W}^t; \boldsymbol{\xi}_n) = \mathbf{A}^t \left( \mathbf{H}_n^t + (\mathbf{H}_n^t)^\intercal \right), \quad n \in \mathcal{N}, t \in \mathcal{T}, \tag{2}$$

where $\mathbf{H}_n^t = \mathbf{L}^\intercal \nabla F_n(\mathbf{W}^t; \boldsymbol{\xi}_n)\mathbf{R}^\intercal \in \mathbb{R}^{k \times k}$. Client $n$ performs stochastic gradient descent to update matrix $\mathbf{A}^t$. We define $\eta$ as the learning rate. Let $\mathbf{A}_n^{t+\frac{1}{2}}$ denote the locally updated low-rank matrix of client $n$ in the $t$-th fine-tuning round, which is given by

$$\mathbf{A}_n^{t+\frac{1}{2}} = \mathbf{A}^t - \eta \nabla_{\mathbf{A}} F_n(\mathbf{W}^t; \boldsymbol{\xi}_n), \quad n \in \mathcal{N}, t \in \mathcal{T}. \tag{3}$$

**(ii) Model Aggregation:** When compared with the conventional federated LoRA schemes, in which clients must upload both locally-updated low-rank matrices, client $n$ in FLoRG only needs to transmit $\mathbf{A}_n^{t+\frac{1}{2}}$ to the central server, thereby reducing the per-round uplink communication overhead by more than half. Let $\mathbf{Q}_n^{t+\frac{1}{2}} = \left( \mathbf{A}_n^{t+\frac{1}{2}} \right)^\intercal \mathbf{A}_n^{t+\frac{1}{2}}$ denote the locally updated Gram matrix for client $n$ in the $t$-th fine-tuning round. After receiving the locally updated parameter matrices from all clients, the central server performs model aggregation with respect to $\mathbf{A}_n^{t+\frac{1}{2}}$ as

$$\mathbf{Q}^{t+1} = \frac{1}{N} \sum_{n \in \mathcal{N}} \mathbf{Q}_n^{t+\frac{1}{2}}, \quad t \in \mathcal{T}, \tag{4}$$

where $\mathbf{Q}^{t+1} \in \mathbb{R}^{k \times k}$ is the aggregated Gram matrix at the end of the $t$-th fine-tuning round. Due to the Gram matrix design, aggregating $\left( \mathbf{A}_n^{t+\frac{1}{2}} \right)^\intercal \mathbf{A}_n^{t+\frac{1}{2}}$ is linear and preserves the positive semi-definite (PSD) property. Therefore, the central server can obtain the true aggregated matrix. This

removes the bilinear inconsistency (i.e., aggregation error) induced by aggregating matrices $\mathbf{B}_n$ and $\mathbf{A}_n$ separately as in conventional federated LoRA schemes. Let $r',t$ denote the rank of matrix $\mathbf{Q}^{t+1}$, which satisfies

$$\text{rank}(\mathbf{Q}_n^{t+\frac{1}{2}}) \leq r',t = \text{rank}(\mathbf{Q}^{t+1}) \overset{(a)}{\leq} \min\{k, Nr\}, \tag{5}$$

where inequality (a) follows from the subadditivity property of rank operator.

**(iii) Decomposition with Procrustes Alignment:** Since $\mathbf{Q}^{t+1}$ is a square PSD Gram matrix, the central server performs eigendecomposition to $\mathbf{Q}^{t+1}$ as follows:

$$\mathbf{Q}^{t+1} = (\mathbf{P}^{t+1})^\mathsf{T}\Lambda^{t+1}\mathbf{P}^{t+1}, \quad t \in \mathcal{T}, \tag{6}$$

where $\mathbf{P}^{t+1} \in \mathbb{R}^{r',t \times k}$. $\Lambda^{t+1} \in \mathbb{R}^{r',t \times r',t}$ denotes the eigenvalue matrix of $\mathbf{Q}^{t+1}$. Thus, a canonical decomposition which satisfies $(\tilde{\mathbf{A}}^{t+1})^\mathsf{T}\tilde{\mathbf{A}}^{t+1} = \mathbf{Q}^{t+1}$ is

$$\tilde{\mathbf{A}}^{t+1} = (\Lambda^{t+1})^{\frac{1}{2}}\mathbf{P}^{t+1}, \quad t \in \mathcal{T}, \tag{7}$$

where $\tilde{\mathbf{A}}^{t+1}$ is a valid decomposition of $\mathbf{Q}^{t+1}$. However, directly applying $\tilde{\mathbf{A}}^{t+1}$ for the subsequent fine-tuning round (i.e., $\mathbf{A}^{t+1} := \tilde{\mathbf{A}}^{t+1}$) may yield sub-optimal performance since the decomposition yields two challenges: **non-unique decomposition** and **rank mismatch**.

First, the above expression of matrix $\tilde{\mathbf{A}}^{t+1}$ provides a canonical representation of the decomposition of $\mathbf{Q}^{t+1}$, which is non-unique. This is because for any matrix with orthogonal columns $\mathbf{O} \in \mathbb{R}^{r',t \times r',t}$, we have $(\mathbf{O}\tilde{\mathbf{A}}^{t+1})^\mathsf{T}\mathbf{O}\tilde{\mathbf{A}}^{t+1} = \mathbf{Q}^{t+1}$. Furthermore, although the decomposition guarantees that $(\tilde{\mathbf{A}}^{t+1})^\mathsf{T}\tilde{\mathbf{A}}^{t+1}$ preserves the aggregated Gram matrix, there exist many such decompositions due to the non-uniqueness of decompositions (e.g., Cholesky decomposition or singular value decomposition (SVD)) when the rank is deficient or the eigenvalues have multiplicities. However, this non-uniqueness affects the future fine-tuning. From the update rule in eqn. (2), each client performs local fine-tuning in the $(t+1)$-th fine-tuning round using the gradient which depends explicitly on $\mathbf{A}^{t+1}$. As a result, different decompositions of $\mathbf{Q}^{t+1}$ may yield different $\mathbf{A}^{t+1}$ which result in divergent gradient paths. These paths may potentially lead to unstable fine-tuning performance across fine-tuning rounds.

Second, as stated in inequality (5), the rank of the decomposed matrix may be different from that of the original one (i.e., $r',t \neq r$). In such cases, we need to recover a matrix of the target rank $r$ for consistency across fine-tuning rounds.

The aforementioned challenges motivate reparameterizing matrix $\tilde{\mathbf{A}}^{t+1}$ to stabilize the subsequent fine-tuning process while enforcing the target rank $r$. To this end, we propose Procrustes alignment to project $\tilde{\mathbf{A}}^{t+1}$ onto the $r$-rank subspace. Let $\mathbf{S}^t \in \mathbb{R}^{r \times r',t}$ denote this Procrustes alignment matrix in the $t$-th fine-tuning round. The matrix after projection is denoted as $\mathbf{S}^t\tilde{\mathbf{A}}^{t+1}$. In particular, Procrustes alignment minimizes the Frobenius norm between the matrix after projection $\mathbf{S}^t\tilde{\mathbf{A}}^{t+1}$ and $\mathbf{A}^t$, which minimizes the drift due to the non-uniqueness of matrix decomposition. We then formulate the following optimization problem in the $t$-th fine-tuning round:

$$\mathcal{P}_1: \quad \underset{\mathbf{S}^t}{\text{minimize}} \quad \left\|\mathbf{S}^t\tilde{\mathbf{A}}^{t+1} - \mathbf{A}^t\right\|_F^2 \tag{8a}$$

$$\text{subject to} \quad (\mathbf{S}^t)^\mathsf{T}\mathbf{S}^t = \mathbf{I}_{r',t}. \tag{8b}$$

To solve problem $\mathcal{P}_1$, we first convert the objective function (8a) into the following form:

$$\left\|\mathbf{S}^t\tilde{\mathbf{A}}^{t+1} - \mathbf{A}^t\right\|_F^2 = \text{Tr}\left((\mathbf{A}^t)^\mathsf{T}\mathbf{A}^t\right) + \text{Tr}\left((\tilde{\mathbf{A}}^{t+1})^\mathsf{T}(\mathbf{S}^t)^\mathsf{T}\mathbf{S}^t\tilde{\mathbf{A}}^{t+1}\right) - 2\,\text{Tr}\left(\mathbf{A}^t(\tilde{\mathbf{A}}^{t+1})^\mathsf{T}(\mathbf{S}^t)^\mathsf{T}\right)$$

$$= \left\|\mathbf{A}^t\right\|_F^2 + \left\|\tilde{\mathbf{A}}^{t+1}\right\|_F^2 - 2\,\text{Tr}\left(\mathbf{A}^t(\tilde{\mathbf{A}}^{t+1})^\mathsf{T}(\mathbf{S}^t)^\mathsf{T}\right). \tag{9}$$

Since $\|\mathbf{A}^t\|_F^2$ and $\|\tilde{\mathbf{A}}^{t+1}\|_F^2$ have been determined after matrix decomposition at the end of the $t$-th fine-tuning round, problem $\mathcal{P}_1$ is equivalent to the following problem:

$$\mathcal{P}_2: \quad \underset{\mathbf{S}^t}{\text{maximize}} \quad \text{Tr}\left(\mathbf{A}^t(\tilde{\mathbf{A}}^{t+1})^\mathsf{T}(\mathbf{S}^t)^\mathsf{T}\right) \tag{10a}$$

$$\text{subject to} \quad \text{constraint (8b).}$$

Then, we present the following theorem to obtain the optimal solution to problems $\mathcal{P}_2$ and $\mathcal{P}_1$, with the proof provided in Appendix A.1.

**Theorem 1.** (Optimal Procrustes Alignment Matrix) *We denote the SVD of matrix $\mathbf{A}^t(\tilde{\mathbf{A}}^{t+1})^\intercal$ as $\mathbf{A}^t(\tilde{\mathbf{A}}^{t+1})^\intercal = \mathbf{U}^t\boldsymbol{\Sigma}^t(\mathbf{V}^t)^\intercal$, where $\mathbf{U}^t \in \mathbb{R}^{r \times r',t}$ and $\mathbf{V}^t \in \mathbb{R}^{r',t \times r',t}$ have orthogonal columns. Let $\boldsymbol{\Sigma}^t = \mathrm{diag}(\sigma_1^t, \sigma_2^t, \ldots, \sigma_{r',t}^t) \in \mathbb{R}^{r',t \times r',t}$ denote the diagonal matrix of singular values of $\mathbf{A}^t(\tilde{\mathbf{A}}^{t+1})^\intercal$. The optimal solution $\mathbf{S}^{t,\star}$ to problems $\mathcal{P}_2$ and $\mathcal{P}_1$ satisfies*
$$\mathbf{S}^{t,\star} = \mathbf{U}^t(\mathbf{V}^t)^\intercal. \tag{11}$$

The main benefits of our proposed Procrustes alignment are two-fold. First, it resolves the issue due to non-unique decomposition. In particular, the Procrustes alignment approach selects, among all valid decompositions of $\mathbf{Q}^{t+1}$, the one which is nearest to that in the last fine-tuning round (i.e., $\mathbf{A}^t$) in Frobenius norm. Thus, it stabilizes the gradients in the subsequent fine-tuning rounds. Second, it addresses the rank mismatch issue of the decomposed matrix by using a semi-orthogonal Procrustes alignment matrix to project $\tilde{\mathbf{A}}^{t+1}$ with rank $r',t$ onto the target $r$-rank subspace.

After the central server has determined matrix $\mathbf{S}^{t,\star}$, it calculates the low-rank matrix for the $(t+1)$-th fine-tuning round as $\mathbf{A}^{t+1} = \mathbf{S}^{t,\star}\tilde{\mathbf{A}}^{t+1}$. Thus, the full model in the $(t+1)$-th fine-tuning round is given by
$$\mathbf{W}^{t+1} = \mathbf{W}^0 + \mathbf{L}(\mathbf{A}^{t+1})^\intercal \mathbf{A}^{t+1}\mathbf{R}. \tag{12}$$

Then, the central server broadcasts the matrix $\mathbf{A}^{t+1}$ to all clients on the downlink. Note that the proposed FLoRG can effectively reduce the per-round communication overhead by more than half when compared with traditional federated LoRA schemes. For a model with $L$ LoRA layers, the total per-round computation overhead of eigendecomposition and Procrustes alignment at the server is $\mathcal{O}(Lk^3) + \mathcal{O}(L(krr',t + \min(r, r',t)^2 \max(r, r',t)))$. From this complexity analysis, the per-round server computation overhead scales linearly with the number of LoRA layers and is independent of the number of clients (beyond the standard Gram aggregation), indicating that FLoRG is scalable even with many layers and many clients. The workflow of our proposed FLoRG is presented in Appendix A.2.

## 3.2 THEORETICAL ANALYSIS

In this section, we analyze the convergence rate of our proposed FLoRG. We define the global loss function in the $t$-th fine-tuning round as $f(\mathbf{W}^t) = \frac{1}{N}\sum_{n \in \mathcal{N}} f_n(\mathbf{W}^t)$, and its gradient as $\nabla f(\mathbf{W}^t) = \frac{1}{N}\sum_{n \in \mathcal{N}} \nabla f_n(\mathbf{W}^t) \in \mathbb{R}^{d_{\text{out}} \times d_{\text{in}}}$. Without loss of generality, we consider nonconvex loss functions in our analysis. We first present the following assumptions which are widely used in the literature (e.g., (Li et al., 2020; Wang et al., 2020; Guo et al., 2025)).

**Assumption 1.** ($L$-Smoothness (Li et al., 2020; Wang et al., 2020)) *The loss function of each client $n$ is continuously differentiable and $L$-smooth. That is, for two arbitrary matrices $\mathbf{W}^t$ and $\mathbf{W}^{t+1}$, we have $f_n(\mathbf{W}^{t+1}) \leq f_n(\mathbf{W}^t) + \langle \nabla f_n(\mathbf{W}^t), \mathbf{W}^{t+1} - \mathbf{W}^t \rangle_F + \frac{L}{2}\|\mathbf{W}^{t+1} - \mathbf{W}^t\|_F^2, t \in \mathcal{T}, n \in \mathcal{N}.$*

**Assumption 2.** (Bounded Gradient (Li et al., 2020; Wang et al., 2020)) *The local stochastic gradient of $\mathbf{A}$ is upper-bounded, i.e., $\mathbb{E}_{\boldsymbol{\xi}_n \sim \mathcal{D}_n}\left[\|\nabla_{\mathbf{A}} F_n(\mathbf{W}^t; \boldsymbol{\xi}_n)\|_F^2\right] \leq \psi, t \in \mathcal{T}, n \in \mathcal{N}.$*

**Assumption 3.** (Bounded Parameter Space (Guo et al., 2025)) *The Frobenius norm of model parameter matrices $\mathbf{A}^t$ and $\tilde{\mathbf{A}}^t$ are upper-bounded by two positive constants $C_{\mathbf{A}}$ and $\tilde{C}_{\mathbf{A}}$, respectively, i.e., $\|\mathbf{A}^t\|_F \leq C_{\mathbf{A}}, \quad \|\tilde{\mathbf{A}}^t\|_F \leq \tilde{C}_{\mathbf{A}}, \quad t \in \mathcal{T}.$*

In addition, we present two lemmas to facilitate our convergence analysis, with the proof provided in Appendices A.3 and A.4, respectively.

**Lemma 1.** *For illustration simplicity, we denote $\mathbf{G}_n^t = \nabla F_n(\mathbf{W}^t; \boldsymbol{\xi}_n) \in \mathbb{R}^{d_{\text{out}} \times d_{\text{in}}}$. We denote $\mathbf{H}^t = \frac{1}{N}\sum_{n \in \mathcal{N}} \mathbf{H}_n^t = \frac{1}{N}\sum_{n \in \mathcal{N}} \mathbf{L}^\intercal \mathbf{G}_n^t \mathbf{R}^\intercal$. Let $\lambda_{\min}(\mathbf{X})$ denote the smallest positive eigenvalue of matrix $\mathbf{X}$. For any matrices $\mathbf{A}^t$ and $\mathbf{H}^t$, we have*

$$\left\langle \mathbf{H}^t, \ (\mathbf{A}^t)^\intercal \mathbf{A}^t(\mathbf{H}^t + (\mathbf{H}^t)^\intercal) + (\mathbf{H}^t + (\mathbf{H}^t)^\intercal)(\mathbf{A}^t)^\intercal \mathbf{A}^t \right\rangle_F \geq 4\lambda_{\min}((\mathbf{A}^t)^\intercal \mathbf{A}^t)\left\|\mathbf{H}^t\right\|_F^2, \ t \in \mathcal{T}. \tag{13}$$

**Lemma 2.** *Let $\mathbf{S}^t$ denote an arbitrary Procrustes alignment matrix in the $t$-th fine-tuning round. Let $\sigma_{\min}(\mathbf{X})$ denote the smallest positive singular value of matrix $\mathbf{X}$. We define $\Delta_{\mathrm{proc}}^{t+1} = \left\|\mathbf{S}^t\tilde{\mathbf{A}}^{t+1} - \mathbf{A}^t\right\|_F^2 - \left\|\mathbf{S}^{t,\star}\tilde{\mathbf{A}}^{t+1} - \mathbf{A}^t\right\|_F^2 \geq 0$. The difference of two Procrustes alignment matrices is bounded as follows:*

$$\left\|\mathbf{S}^t - \mathbf{S}^{t,\star}\right\|_F^2 \leq \frac{\Delta_{\mathrm{proc}}^{t+1}}{\sigma_{\min}\left(\tilde{\mathbf{A}}^{t+1}(\mathbf{A}^t)^\intercal\right)}, \quad t \in \mathcal{T}. \tag{14}$$

Now, we present the convergence rate of our proposed FLoRG in the following theorem, with the proof provided in Appendix A.5.

**Theorem 2.** (Convergence Rate of FLoRG) *We denote the optimal model as $\mathbf{W}^\star$. Under Assumptions $1-3$ and Lemmas $1-2$, if the learning rate satisfies $\eta < 4\min_{t\in\mathcal{T}}\{\lambda_{\min}((\mathbf{A}^t)^\intercal\mathbf{A}^t)\}$, the convergence rate of our proposed FLoRG is bounded by*

$$\frac{1}{T}\sum_{t\in\mathcal{T}}\left\|\nabla f(\mathbf{W}^t)\right\|_F^2 \leq \underbrace{\frac{f(\mathbf{W}^1) - f(\mathbf{W}^\star)}{T\Omega}}_{\text{Initial optimality gap}} + \underbrace{\frac{\eta^2\psi^2}{2\Omega} + \frac{3L\eta^2\psi\left(\eta^2\psi + 2C_{\mathbf{A}}^2\right)}{2\Omega}}_{\text{Residual bias term}}$$

$$+ \underbrace{\frac{\eta\psi\tilde{C}_{\mathbf{A}}^2\sum_{t\in\mathcal{T}}\frac{\Delta_{\mathrm{proc}}^{t+1}}{\sigma_{\min}(\tilde{\mathbf{A}}^{t+1}(\mathbf{A}^t)^\intercal)}}{2NT\Omega\min_{t\in\mathcal{T}}\{\lambda_{\min}((\mathbf{A}^t)^\intercal\mathbf{A}^t)\}}}_{\text{Procrustes alignment drift term}}, \tag{15}$$

*where $\Omega = 2\eta\min_{t\in\mathcal{T}}\{\lambda_{\min}((\mathbf{A}^t)^\intercal\mathbf{A}^t)\} - \frac{\eta^2}{2}$.*

Based on the theoretical analysis, we can observe that the convergence rate of our proposed FLoRG depends on three terms. The first term is the initial optimality gap, which diminishes as the number of fine-tuning rounds $T$ increases. The second term is the non-diminishing residual bias term. The third term is the Procrustes alignment drift term, which captures the impact of the Procrustes alignment on the convergence rate. When the Procrustes alignment is applied, $\Delta_{\mathrm{proc}}^{t+1}$ becomes zero. Hence, the third term becomes zero, under which we can achieve a tighter bound and improve the convergence rate.

## 4 Experimental Results

### 4.1 Experiment Setup

**Base Models, Datasets, and Baseline Schemes:** We choose OPT-125M (Zhang et al., 2022), RoBERTa-large (Liu et al., 2019), and Llama-3.2-3B (Grattafiori et al., 2024) as three base models with three different scales. In particular, OPT-125M, RoBERTa-large, and Llama-3.2-3B have 125M, 355M, and 3B model parameters. We choose GLUE (Wang et al., 2018) as a natural language understanding benchmark with MRPC, QQP, MNLI, QNLI, WNLI, and RTE datasets. In addition, we choose SQuAD (Rajpurkar et al., 2016) v1.1 as a question-answering dataset. We compare the performance of our proposed FLoRG with the following baseline schemes:

- **FedIT** (Zhang et al., 2024): Client $n \in \mathcal{N}$ transmits the locally updated LoRA matrices $\mathbf{B}_n$ and $\mathbf{A}_n$ to the central server. The central server aggregates $\mathbf{B}_n$ and $\mathbf{A}_n$ separately.
- **FeDeRA** (Yan et al., 2024): Client $n \in \mathcal{N}$ transmits locally updated LoRA matrices $\mathbf{B}_n$ and $\mathbf{A}_n$ to the central server. The central server aggregates $\mathbf{B}_n\mathbf{A}_n$ and performs SVD to obtain the updated matrices $\mathbf{B}_n$ and $\mathbf{A}_n$.
- **FFA-LoRA** (Sun et al., 2024): Client $n \in \mathcal{N}$ freezes matrix $\mathbf{A}_n$ and only updates matrix $\mathbf{B}_n$. The central server performs aggregation on matrix $\mathbf{B}_n$.
- **FedSA-LoRA** (Guo et al., 2025): Client $n \in \mathcal{N}$ locally updates matrices $\mathbf{B}_n$ and $\mathbf{A}_n$ but only shares matrix $\mathbf{A}_n$ for aggregation.
- **FedEx-LoRA** (Singhal et al., 2025): Client $n \in \mathcal{N}$ transmits locally updated LoRA matrices $\mathbf{B}_n$ and $\mathbf{A}_n$ to the central server. The central server aggregates $\mathbf{B}_n$ and $\mathbf{A}_n$ separately and determines a residual matrix to mitigate the aggregation error.

**Implementation Details:** To show the learning performance of GLUE datasets, we show the testing accuracy. To characterize the communication overhead, we present the total number of parameters transmitted between all clients and the central server. For GLUE, to model the data heterogeneity across clients' local datasets, we use the Dirichlet distribution $\mathrm{Dir}(\rho)$ to create non-independent and identically distributed (non-IID) data partitioning. In particular, $\rho > 0$ controls the degree of non-IIDness across clients' local datasets. A lower value of $\rho$ indicates a higher degree of data heterogeneity. To show the learning performance of the SQuAD v1.1 dataset, we show the exact match (EM) and F1 scores. For SQuAD v1.1, we consider the dataset to be independently and identically distributed (IID). In the ablation studies, we choose RoBERTa-large as the base model. Unless stated otherwise, we consider all clients participate in the fine-tuning and set $\eta = 5e-5$, $\rho = 0.5$, $N = 20$, and $r = 4$. The details of the hyperparameter settings are presented in Appendix A.6.

## 4.2 LEARNING PERFORMANCE

In this section, we compare the testing accuracy of different schemes under different base models and datasets. Due to the space limit, we present the results under MNLI, QNLI, WNLI, and RTE datasets. The experiments are repeated under 2 random seeds, and we report the average testing accuracy. Results in Table 1 show that our proposed FLoRG outperforms the baseline schemes under those four datasets in most cases. In particular, on OPT-125M, FLoRG improves the testing accuracy over the strongest baseline by 1.52 on MNLI, 1.13 on WNLI, and 0.65 on RTE. On RoBERTa-large, the margins are 0.31 on MNLI, 0.45 on QNLI, 0.37 on WNLI, and 0.28 on RTE. On Llama-3.2-3B, FLoRG improves the testing accuracy over the strongest baseline by 0.41 on MNLI, 0.84 on WNLI, and 0.69 on RTE. Additional experimental results are presented in Appendix A.7. These results validate the superiority of FLoRG.

Table 1: Comparison of the testing accuracy across different baseline schemes.

| Base Model | Dataset | FLoRG | FedIT | FeDeRA | FFA-LoRA | FedSA-LoRA | FedEx-LoRA |
|---|---|---|---|---|---|---|---|
| OPT-125M | MNLI | **87.35** | 79.42 | 81.15 | 83.54 | 84.61 | 85.83 |
| | QNLI | 89.52 | 84.18 | 86.71 | 87.93 | 88.69 | **89.88** |
| | WNLI | **65.28** | 58.45 | 59.34 | 62.61 | 62.83 | 64.15 |
| | RTE | **68.92** | 61.08 | 64.51 | 66.02 | 67.39 | 68.27 |
| RoBERTa-large | MNLI | **91.27** | 84.91 | 88.06 | 89.28 | 90.75 | 90.96 |
| | QNLI | **92.58** | 87.49 | 90.07 | 90.96 | 91.54 | 92.13 |
| | WNLI | **66.48** | 59.34 | 61.83 | 63.72 | 63.47 | 66.11 |
| | RTE | **71.26** | 64.25 | 67.12 | 68.49 | 69.93 | 70.98 |
| Llama-3.2-3B | MNLI | **93.15** | 87.24 | 89.83 | 91.05 | 92.38 | 92.74 |
| | QNLI | 93.12 | 89.17 | 91.45 | 92.53 | **93.27** | 93.05 |
| | WNLI | **68.73** | 61.52 | 64.19 | 65.97 | 66.81 | 67.89 |
| | RTE | **73.84** | 67.08 | 69.75 | 71.33 | 72.56 | 73.15 |

## 4.3 COMPARISON OF THE COMMUNICATION OVERHEAD

In this section, we compare the communication overhead incurred under different baseline schemes. We use the QNLI dataset to conduct the experiments. Results in Table 2 show that to achieve the target test accuracy, our proposed FLoRG uses a much lower total number of transmitted model parameters when compared with the baselines. This demonstrates that FLoRG can significantly reduce the communication overhead by up to $2041\times$ when compared with the baseline schemes. Additional results of the computation overhead and memory usage are presented in Appendix A.7.

## 4.4 ABLATION STUDIES

**Impact of the Procrustes alignment** In this subsection, we study the impact of our proposed Procrustes alignment on the learning performance. Results in Table 3 show that by applying Procrustes

Table 2: Comparison of the total number of transmitted parameters (upload and download) to achieve the target accuracy. Symbol "−" means that the target accuracy cannot be achieved.

| Base Model | Target Acc. | FLoRG | FedIT | FeDeRA | FFA-LoRA | FedSA-LoRA | FedEx-LoRA |
|---|---|---|---|---|---|---|---|
| OPT-125M | 80.00 | $\mathbf{8.2 \times 10^6}$ | $3.78 \times 10^7$ | $2.46 \times 10^7$ | $1.59 \times 10^7$ | $2.10 \times 10^7$ | $1.25 \times 10^{10}$ |
| | 85.00 | $\mathbf{1.07 \times 10^7}$ | − | $4.2 \times 10^7$ | $2.75 \times 10^7$ | $3.61 \times 10^7$ | $1.77 \times 10^{10}$ |
| RoBERTa-large | 80.00 | $\mathbf{8.7 \times 10^6}$ | $4.68 \times 10^7$ | $3.02 \times 10^7$ | $1.85 \times 10^7$ | $2.59 \times 10^7$ | $2.08 \times 10^{10}$ |
| | 85.00 | $\mathbf{1.45 \times 10^7}$ | $8.12 \times 10^7$ | $5.17 \times 10^7$ | $3.35 \times 10^7$ | $4.42 \times 10^7$ | $2.96 \times 10^{10}$ |

alignment, our proposed FLoRG yields a consistent improvement in terms of the testing accuracy. On OPT-125M, Procrustes alignment provides an improvement of 3.40 on MRPC, 2.86 on QQP, 6.27 on MNLI, 2.97 on QNLI, 5.60 on WNLI, and 4.45 on RTE, respectively. On RoBERTa-large, Procrustes alignment provides an improvement of 3.37 on MRPC, 2.84 on QQP, 2.46 on MNLI, 3.86 on QNLI, 4.34 on WNLI, and 4.31 on RTE, respectively, whereas FLoRG without Procrustes alignment can only achieve a comparable testing accuracy to FeDeRA, as shown in Table 1. It showcases the importance of our proposed Procrustes alignment to improve the fine-tuning performance.

Table 3: Comparison of the testing accuracy of FLoRG with and without Procrustes alignment.

| Base Model | FLoRG | MRPC | QQP | MNLI | QNLI | WNLI | RTE |
|---|---|---|---|---|---|---|---|
| OPT-125M | w/ Procrustes alignment | **86.54** | **88.71** | **87.20** | **89.69** | **65.41** | **68.77** |
| | w/o Procrustes alignment | 83.14 | 85.85 | 80.93 | 86.72 | 59.81 | 64.32 |
| RoBERTa-large | w/ Procrustes alignment | **89.87** | **91.27** | **91.39** | **92.48** | **66.41** | **71.40** |
| | w/o Procrustes alignment | 86.50 | 88.43 | 88.93 | 88.62 | 62.07 | 67.09 |

**Impact of the Rank**  In this subsection, we vary $r$ to demonstrate the impact of rank on the fine-tuning performance. In particular, we conduct experiments under $r = 2, 4, 8$, respectively. We present the results under the WNLI and RTE datasets. Results in Table 4 show that our proposed FLoRG outperforms the baseline schemes under different rank settings, demonstrating the robustness of our proposed FLoRG under various ranks. Additional experimental results can be found in Appendix A.7.

Table 4: Comparison of the testing accuracy under different ranks.

| Rank | Dataset | FLoRG | FedIT | FeDeRA | FFA-LoRA | FedSA-LoRA | FedEx-LoRA |
|---|---|---|---|---|---|---|---|
| $r = 2$ | WNLI | **60.55** | 55.57 | 56.30 | 57.50 | 59.14 | 58.32 |
| | RTE | **65.82** | 58.41 | 61.19 | 62.88 | 64.30 | 62.79 |
| $r = 4$ | WNLI | **66.34** | 59.19 | 61.97 | 63.55 | 63.61 | 66.12 |
| | RTE | **71.41** | 64.11 | 66.97 | 68.62 | 70.10 | 71.28 |
| $r = 8$ | WNLI | **68.83** | 61.70 | 63.52 | 65.10 | 66.47 | 68.02 |
| | RTE | **72.10** | 64.78 | 66.99 | 68.02 | 70.61 | 71.30 |

**Robustness to the Data Heterogeneity**  In this subsection, we study the impact of the degree of data heterogeneity across clients' local datasets on the fine-tuning performance. We present the results under the WNLI and RTE datasets. It can be observed in Table 5 that under different degrees of data heterogeneity, our proposed FLoRG outperforms the baseline schemes under more heterogeneous settings. In addition, as the degree of data heterogeneity increases (i.e., $\rho$ decreases), the improvement over the baseline schemes also increases, showcasing the robustness and superiority of our proposed FLoRG under heterogeneous data settings. Additional experimental results are presented in Appendix A.7.

**Matrix Initialization for Matrices $\mathbf{L}$ and $\mathbf{R}$**  In this subsection, we show the impact of the initialization of matrices $\mathbf{L}$ and $\mathbf{R}$ on the learning performance. In particular, we compare our proposed

Table 5: Comparison of the testing accuracy under different degrees of data heterogeneity.

| Non-IIDness | Dataset | FLoRG | FedIT | FeDeRA | FFA-LoRA | FedSA-LoRA | FedEx-LoRA |
|---|---|---|---|---|---|---|---|
| $\rho = 0.1$ | WNLI | **60.12** | 53.07 | 54.21 | 56.14 | 57.74 | 58.84 |
| | RTE | **65.30** | 55.60 | 59.19 | 61.20 | 60.75 | 61.67 |
| $\rho = 0.5$ | WNLI | **66.34** | 59.19 | 61.97 | 63.55 | 63.61 | 66.12 |
| | RTE | **71.41** | 64.11 | 66.97 | 68.62 | 70.10 | 71.28 |
| $\rho = 1$ | WNLI | 67.83 | 61.70 | 63.52 | 64.33 | 65.61 | **68.31** |
| | RTE | 72.21 | 66.90 | 68.71 | 70.40 | 71.78 | **72.55** |

semi-orthogonal initialization with Kaiming initialization (He et al., 2015) and SVD initialization (Boutsidis & Gallopoulos, 2007). Results in Table 6 show that the semi-orthogonal approach outperforms the other two approaches in most cases, demonstrating the effectiveness of our proposed initialization approach for matrices $\mathbf{L}$ and $\mathbf{R}$.

Table 6: Comparison of the testing accuracy under different initialization schemes.

| Base Model | Initialization | MRPC | QQP | MNLI | QNLI | WNLI | RTE |
|---|---|---|---|---|---|---|---|
| OPT-125M | Semi-orthogonal | **86.54** | 88.71 | **87.20** | **89.69** | 65.41 | 68.77 |
| | Kaiming | 84.35 | **88.90** | 85.23 | 87.73 | 62.29 | **69.31** |
| | SVD | 86.41 | 87.69 | 83.19 | 88.74 | 64.37 | 67.69 |
| RoBERTa-large | Semi-orthogonal | **89.87** | **91.27** | 91.39 | **92.48** | 66.41 | **71.40** |
| | Kaiming | 87.68 | 92.34 | 89.11 | 91.57 | 64.19 | 70.32 |
| | SVD | 88.70 | 90.37 | **91.49** | 91.45 | 65.08 | **71.40** |

**Impact of the Client Availability**    In this subsection, we show the impact of the client availability on the learning performance. In particular, we conduct experiments under varying client participation ratios. Results in Table 7 show that under different ratios of client participation, our proposed FLoRG outperforms the baseline schemes in most cases, demonstrating its applicability.

Table 7: Comparison of the testing accuracy under different client participation ratios.

| Client participation ratio | Dataset | FLoRG | FedIT | FeDeRA | FFA-LoRA | FedSA-LoRA | FedEx-LoRA |
|---|---|---|---|---|---|---|---|
| 0.2 | WNLI | **58.42** | 54.15 | 55.20 | 56.30 | 57.10 | 56.46 |
| | RTE | **63.25** | 57.80 | 59.45 | 60.92 | 61.85 | 61.35 |
| 0.5 | WNLI | **64.50** | 58.10 | 60.35 | 61.80 | 62.40 | 63.58 |
| | RTE | **69.15** | 62.50 | 65.20 | 66.75 | 68.20 | 68.61 |
| 1 | WNLI | **66.34** | 59.19 | 61.97 | 63.55 | 63.61 | 66.12 |
| | RTE | **71.41** | 64.11 | 66.97 | 68.62 | 70.10 | 71.28 |

## 5    CONCLUSION

In this work, we proposed a federated fine-tuning framework with low-rank Gram matrices called FLoRG. In particular, FLoRG features a single low-rank matrix instead of two low-rank matrices as in the conventional LoRA module. By transmitting this matrix and aggregating the corresponding Gram matrix, FLoRG eliminates the error induced by separately aggregating two matrices and significantly reduces the per-round communication overhead. Moreover, we proposed a Procrustes alignment approach to reparameterize the decomposed low-rank matrix after model aggregation. We theoretically analyzed the convergence rate of our proposed FLoRG framework and characterized the impact of our proposed Procrustes alignment on the convergence. Experimental results show that our proposed FLoRG framework achieves a higher testing accuracy and can reduce the communication overhead by up to $2041\times$ when compared with five baseline schemes.

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

# A APPENDIX

## A.1 PROOF OF THEOREM 1

During the Procrustes alignment, we define $\mathbf{Z}^t = (\mathbf{U}^t)^\intercal \mathbf{S}^t \mathbf{V}^t \in \mathbb{R}^{r',t \times r',t}$. Therefore, $\mathbf{S}^t$ satisfies

$$\mathbf{S}^t = \mathbf{U}^t \mathbf{Z}^t (\mathbf{V}^t)^\intercal. \tag{16}$$

Note that $(\mathbf{S}^t)^\intercal \mathbf{S}^t = \mathbf{I}_{r',t}$. In addition, since matrices $\mathbf{U}^t$ and $\mathbf{V}^t$ have orthogonal columns, we have

$$\begin{aligned}
\operatorname{Tr}\left(\mathbf{A}^t(\tilde{\mathbf{A}}^{t+1})^\intercal (\mathbf{S}^t)^\intercal\right) &= \operatorname{Tr}\left(\mathbf{U}^t \mathbf{\Sigma}^t (\mathbf{V}^t)^\intercal \mathbf{V}^t (\mathbf{Z}^t)^\intercal (\mathbf{U}^t)^\intercal\right) \\
&= \operatorname{Tr}\left(\mathbf{\Sigma}^t (\mathbf{Z}^t)^\intercal\right) \\
&= \sum_{j=1}^{r',t} \sigma_j^t z_{j,j}^t.
\end{aligned} \tag{17}$$

Since matrix $\mathbf{Z}^t$ satisfies $(\mathbf{Z}^t)^\intercal \mathbf{Z}^t = \mathbf{I}_{r',t}$, each element $z_{j,j}^t$ yields $z_{j,j}^t \leq 1$, we have $\operatorname{Tr}\left(\mathbf{A}^t(\tilde{\mathbf{A}}^{t+1})^\intercal (\mathbf{S}^t)^\intercal\right) \leq \sum_{j=1}^{r',t} \sigma_j^t$, with the equality achieved at $\mathbf{Z}^t = \mathbf{I}_{r',t}$. Therefore, when the maximum objective is achieved, matrix $\mathbf{S}^{t,\star}$ satisfies

$$\mathbf{S}^{t,\star} = \mathbf{U}^t \mathbf{I}_{r',t} (\mathbf{V}^t)^\intercal = \mathbf{U}^t (\mathbf{V}^t)^\intercal. \tag{18}$$

This completes the proof of Theorem 1.

## A.2 ALGORITHM FOR FLoRG

---

**Algorithm 1** FLoRG

---

1: **Input**: Local fine-tuning datasets $\mathcal{D}_n$, $n \in \mathcal{N}$; learning rate $\eta$; pretrained model $\mathbf{W}^0$; initialized low-rank matrix $\mathbf{A}^1$.
2: The central server initializes $\mathbf{L}$ and $\mathbf{R}$ with orthogonal columns and broadcasts them to all clients.
3: $\mathbf{W}^1 := \mathbf{W}^0 + \mathbf{L}(\mathbf{A}^1)^\intercal \mathbf{A}^1 \mathbf{R}$.
4: **For** $t \in \mathcal{T}$ **do**
5:     The central server broadcasts $\mathbf{A}^t$ to all clients.
6:     **For** client $n \in \mathcal{N}$ **in parallel do**
7:         $\mathbf{A}_n^{t+\frac{1}{2}} := \mathbf{A}^t - \eta \nabla_{\mathbf{A}} F_n(\mathbf{W}^t; \boldsymbol{\xi}_n)$.
8:         Transmits $\mathbf{A}_n^{t+\frac{1}{2}}$ to the central server.
9:     **End for**
10:     $\mathbf{Q}^{t+1} := \frac{1}{N} \sum_{n \in \mathcal{N}} \left(\mathbf{A}_n^{t+\frac{1}{2}}\right)^\intercal \mathbf{A}_n^{t+\frac{1}{2}}$.
11:     Perform eigendecomposition to $\mathbf{Q}^{t+1}$ and obtain $\tilde{\mathbf{A}}^{t+1} := (\Lambda^{t+1})^{\frac{1}{2}} \mathbf{P}^{t+1}$.
12:     Perform Procrustes alignment to $\tilde{\mathbf{A}}^{t+1}$ and obtain $\mathbf{S}^{t,\star} := \mathbf{U}^t (\mathbf{V}^t)^\intercal$.
13:     $\mathbf{A}^{t+1} := \mathbf{S}^{t,\star} \tilde{\mathbf{A}}^{t+1}$.
14:     $\mathbf{W}^{t+1} := \mathbf{W}^0 + \mathbf{L}(\mathbf{A}^{t+1})^\intercal \mathbf{A}^{t+1} \mathbf{R}$.
15: **End for**
16: **Output**: Fine-tuned model $\mathbf{W}^{T+1}$.

---

## A.3 PROOF OF LEMMA 1

$$\begin{aligned}
&\left\langle \mathbf{H}^t, \ (\mathbf{A}^t)^\intercal \mathbf{A}^t \left(\mathbf{H}^t + (\mathbf{H}^t)^\intercal\right) + \left(\mathbf{H}^t + (\mathbf{H}^t)^\intercal\right)(\mathbf{A}^t)^\intercal \mathbf{A}^t \right\rangle_F \\
&\overset{(a)}{=} \left\langle \tfrac{\mathbf{H}^t + (\mathbf{H}^t)^\intercal}{2}, \ (\mathbf{A}^t)^\intercal \mathbf{A}^t \left(\mathbf{H}^t + (\mathbf{H}^t)^\intercal\right) + \left(\mathbf{H}^t + (\mathbf{H}^t)^\intercal\right)(\mathbf{A}^t)^\intercal \mathbf{A}^t \right\rangle_F \\
&= 2\left\langle \tfrac{\mathbf{H}^t + (\mathbf{H}^t)^\intercal}{2}, \ (\mathbf{A}^t)^\intercal \mathbf{A}^t \tfrac{\mathbf{H}^t + (\mathbf{H}^t)^\intercal}{2} \right\rangle_F + 2\left\langle \tfrac{\mathbf{H}^t + (\mathbf{H}^t)^\intercal}{2}, \ \tfrac{\mathbf{H}^t + (\mathbf{H}^t)^\intercal}{2}(\mathbf{A}^t)^\intercal \mathbf{A}^t \right\rangle_F \\
&\overset{(b)}{\geq} 2\,\lambda_{\min}((\mathbf{A}^t)^\intercal \mathbf{A}^t)\left\|\tfrac{\mathbf{H}^t + (\mathbf{H}^t)^\intercal}{2}\right\|_F^2 + 2\,\lambda_{\min}((\mathbf{A}^t)^\intercal \mathbf{A}^t)\left\|\tfrac{\mathbf{H}^t + (\mathbf{H}^t)^\intercal}{2}\right\|_F^2
\end{aligned}$$

$$= 4\lambda_{\min}\left((\mathbf{A}^t)^\mathsf{T}\mathbf{A}^t\right)\left\|\tfrac{\mathbf{H}^t+(\mathbf{H}^t)^\mathsf{T}}{2}\right\|_F^2$$

$$= 4\lambda_{\min}\left((\mathbf{A}^t)^\mathsf{T}\mathbf{A}^t\right)\left\|\mathbf{H}^t\right\|_F^2, \tag{19}$$

where equality (a) is obtained due to the fact that the second factor within the inner product is symmetric. Thus, the skew-symmetric part of $\mathbf{H}^t$ is orthogonal. Inequality (b) results from $(\mathbf{A}^t)^\mathsf{T}\mathbf{A}^t \succeq \lambda_{\min}\left((\mathbf{A}^t)^\mathsf{T}\mathbf{A}^t\right)\mathbf{I}_k$.

This completes the proof of Lemma 1.

### A.4 PROOF OF LEMMA 2

We expand $\left\|\mathbf{S}^t\tilde{\mathbf{A}}^{t+1} - \mathbf{A}^t\right\|_F^2$ as follows:

$$\left\|\mathbf{S}^t\tilde{\mathbf{A}}^{t+1} - \mathbf{A}^t\right\|_F^2 = \left\|\tilde{\mathbf{A}}^{t+1}\right\|_F^2 + \left\|\mathbf{A}^t\right\|_F^2 - 2\,\mathrm{Tr}\left(\mathbf{S}^t\tilde{\mathbf{A}}^{t+1}(\mathbf{A}^t)^\mathsf{T}\right). \tag{20}$$

Similarly, we have

$$\left\|\mathbf{S}^{t,\star}\tilde{\mathbf{A}}^{t+1} - \mathbf{A}^t\right\|_F^2 = \left\|\tilde{\mathbf{A}}^{t+1}\right\|_F^2 + \left\|\mathbf{A}^t\right\|_F^2 - 2\,\mathrm{Tr}\left(\mathbf{S}^{t,\star}\tilde{\mathbf{A}}^{t+1}(\mathbf{A}^t)^\mathsf{T}\right). \tag{21}$$

Therefore, $\Delta_{\mathrm{proc}}^{t+1}$ satisfies

$$\Delta_{\mathrm{proc}}^{t+1} = 2\left(\mathrm{Tr}\left(\mathbf{S}^{t,\star}\tilde{\mathbf{A}}^{t+1}(\mathbf{A}^t)^\mathsf{T}\right) - \mathrm{Tr}\left(\mathbf{S}^t\tilde{\mathbf{A}}^{t+1}(\mathbf{A}^t)^\mathsf{T}\right)\right). \tag{22}$$

Let $\mathbf{M}^t = \tilde{\mathbf{A}}^{t+1}(\mathbf{A}^t)^\mathsf{T} = \mathbf{U}_{\mathbf{M}}^t\mathbf{\Sigma}_{\mathbf{M}}^t\mathbf{V}_{\mathbf{M}}^t$, where matrices $\mathbf{U}_{\mathbf{M}}^t \in \mathbb{R}^{r',t \times r',t}$ and $\mathbf{V}_{\mathbf{M}}^t \in \mathbb{R}^{r',t \times r}$ have orthogonal columns, and $\mathbf{\Sigma}_{\mathbf{M}}^t \in \mathbb{R}^{r',t \times r',t}$. Let $\mathbf{P}^t = \mathbf{U}_{\mathbf{M}}^t(\mathbf{S}^t)^\mathsf{T}(\mathbf{V}_{\mathbf{M}}^t)^\mathsf{T}$ and $\mathbf{P}^{t,\star} = \mathbf{U}_{\mathbf{M}}^t(\mathbf{S}^{t,\star})^\mathsf{T}(\mathbf{V}_{\mathbf{M}}^t)^\mathsf{T} = \mathbf{I}_{r',t}$. Then, we have

$$\mathrm{Tr}\left(\mathbf{S}^t\tilde{\mathbf{A}}^{t+1}(\mathbf{A}^t)^\mathsf{T}\right) = \mathrm{Tr}\left(\mathbf{P}^t\mathbf{\Sigma}_{\mathbf{M}}^t\right) = \sum_{j=1}^{r',t}\sigma_j^t p_{j,j}^t. \tag{23}$$

Similarly, we have

$$\mathrm{Tr}\left(\mathbf{S}^{t,\star}\tilde{\mathbf{A}}^{t+1}(\mathbf{A}^t)^\mathsf{T}\right) = \sum_{j=1}^{r',t}\sigma_j^t. \tag{24}$$

By combining eqns. (22), (23), and (24), we obtain

$$\Delta_{\mathrm{proc}}^{t+1} = 2\sum_{j=1}^{r',t}\sigma_j^t(1 - p_{j,j}^t). \tag{25}$$

Since matrices $\mathbf{U}_{\mathbf{M}}^t$ and $\mathbf{V}_{\mathbf{M}}^t$ have orthogonal columns, we have

$$\begin{aligned}
\left\|\mathbf{S}^t - \mathbf{S}^{t,\star}\right\|_F^2 &= \left\|\mathbf{P}^t - \mathbf{I}_{r',t}\right\|_F^2 \\
&= \mathrm{Tr}\left((\mathbf{P}^t - \mathbf{I}_{r',t})^\mathsf{T}(\mathbf{P}^t - \mathbf{I}_{r',t})\right) \\
&= 2r',t - \mathrm{Tr}(\mathbf{P}^t) - \mathrm{Tr}\left((\mathbf{P}^t)^\mathsf{T}\right) \\
&= 2\sum_{j=1}^{r',t}(1 - p_{j,j}^t).
\end{aligned} \tag{26}$$

By combining eqns. (25) and (26), we have

$$\Delta_{\mathrm{proc}}^{t+1} = 2\sum_{j=1}^{r',t}\sigma_j^t(1 - p_{j,j}^t)$$

$$\overset{(a)}{\geq} 2\sigma_{\min}\left(\tilde{\mathbf{A}}^{t+1}(\mathbf{A}^t)^{\intercal}\right)\sum_{j=1}^{r',t}(1-p_{j,j}^t)$$

$$= \sigma_{\min}\left(\tilde{\mathbf{A}}^{t+1}(\mathbf{A}^t)^{\intercal}\right)\left\|\mathbf{P}^t-\mathbf{I}_{r',t}\right\|_F^2$$

$$= \sigma_{\min}\left(\tilde{\mathbf{A}}^{t+1}(\mathbf{A}^t)^{\intercal}\right)\left\|\mathbf{S}^t-\mathbf{S}^{t,\star}\right\|_F^2, \tag{27}$$

where inequality (a) is obtained due to the fact that $\sigma_{\min}\left(\tilde{\mathbf{A}}^{t+1}(\mathbf{A}^t)^{\intercal}\right) \leq \sigma_j^t, 1 \leq j \leq r',t$. By rearranging this inequality, we have

$$\left\|\mathbf{S}^t-\mathbf{S}^{t,\star}\right\|_F^2 \leq \frac{\Delta_{\mathrm{proc}}^{t+1}}{\sigma_{\min}\left(\tilde{\mathbf{A}}^{t+1}(\mathbf{A}^t)^{\intercal}\right)}. \tag{28}$$

This completes the proof of Lemma 2.

### A.5 PROOF OF THEOREM 2

Based on Assumption 1, we expand $\mathbb{E}[f(\mathbf{W}^{t+1})]$ as

$$\mathbb{E}\left[f(\mathbf{W}^{t+1})\right] \leq \mathbb{E}\left[f(\mathbf{W}^t)\right] + \underbrace{\mathbb{E}\left[\langle\nabla f(\mathbf{W}^t), \mathbf{W}^{t+1}-\mathbf{W}^t\rangle_F\right]}_{T_1} + \frac{L}{2}\underbrace{\mathbb{E}\left[\left\|\mathbf{W}^{t+1}-\mathbf{W}^t\right\|_F^2\right]}_{T_2}. \tag{29}$$

We define $\bar{\mathbf{G}}_{\mathbf{A}}^t = \frac{1}{N}\sum_{n\in\mathcal{N}}\nabla_{\mathbf{A}}F_n(\mathbf{W}^t;\boldsymbol{\xi}_n) = \mathbf{A}^t\left(\mathbf{H}^t+(\mathbf{H}^t)^{\intercal}\right)$. $\mathbf{W}^{t+1}-\mathbf{W}^t$ satisfies

$$\mathbf{W}^{t+1}-\mathbf{W}^t$$
$$= \mathbf{W}^0 + \mathbf{L}(\mathbf{A}^{t+1})^{\intercal}\mathbf{A}^{t+1}\mathbf{R} - \left(\mathbf{W}^0 + \mathbf{L}(\mathbf{A}^t)^{\intercal}\mathbf{A}^t\mathbf{R}\right)$$
$$= \mathbf{L}(\mathbf{A}^{t+1})^{\intercal}\mathbf{A}^{t+1}\mathbf{R} - \mathbf{L}(\mathbf{A}^t)^{\intercal}\mathbf{A}^t\mathbf{R}$$
$$= \frac{1}{N}\sum_{n\in\mathcal{N}}\left(\mathbf{L}(\mathbf{A}^{t+1})^{\intercal}\mathbf{A}^{t+1}\mathbf{R} - \mathbf{L}(\mathbf{A}^t)^{\intercal}\mathbf{A}^t\mathbf{R}\right)$$
$$\overset{(a)}{=} \mathbf{L}\frac{1}{N}\sum_{n\in\mathcal{N}}\left((\mathbf{A}_n^{t+\frac{1}{2}})^{\intercal}\mathbf{A}_n^{t+\frac{1}{2}} - (\mathbf{A}^t)^{\intercal}\mathbf{A}^t\right)\mathbf{R}$$
$$= \mathbf{L}\frac{1}{N}\sum_{n\in\mathcal{N}}\left(\left(\mathbf{A}^t-\eta\nabla_{\mathbf{A}}F_n(\mathbf{W}^t;\boldsymbol{\xi}_n)\right)^{\intercal}\left(\mathbf{A}^t-\eta\nabla_{\mathbf{A}}F_n(\mathbf{W}^t;\boldsymbol{\xi}_n)\right) - (\mathbf{A}^t)^{\intercal}\mathbf{A}^t\right)\mathbf{R}$$
$$= \mathbf{L}\left((\mathbf{A}^t)^{\intercal}\mathbf{A}^t - \eta\left((\mathbf{A}^t)^{\intercal}\bar{\mathbf{G}}_{\mathbf{A}}^t + (\bar{\mathbf{G}}_{\mathbf{A}}^t)^{\intercal}\mathbf{A}^t\right) + \eta^2(\bar{\mathbf{G}}_{\mathbf{A}}^t)^{\intercal}\bar{\mathbf{G}}_{\mathbf{A}}^t - (\mathbf{A}^t)^{\intercal}\mathbf{A}^t\right)\mathbf{R}$$
$$= \mathbf{L}\left(\eta^2(\bar{\mathbf{G}}_{\mathbf{A}}^t)^{\intercal}\bar{\mathbf{G}}_{\mathbf{A}}^t - \eta\left((\mathbf{A}^t)^{\intercal}\bar{\mathbf{G}}_{\mathbf{A}}^t + (\bar{\mathbf{G}}_{\mathbf{A}}^t)^{\intercal}\mathbf{A}^t\right)\right)\mathbf{R}, \tag{30}$$

where equality (a) is obtained due to the fact that $\frac{1}{N}\sum_{n\in\mathcal{N}}(\mathbf{A}_n^{t+\frac{1}{2}})^{\intercal}\mathbf{A}_n^{t+\frac{1}{2}} = \mathbf{Q}^{t+1} = (\tilde{\mathbf{A}}^{t+1})^{\intercal}\tilde{\mathbf{A}}^{t+1} = (\mathbf{S}^{t,\star}\tilde{\mathbf{A}}^{t+1})^{\intercal}\mathbf{S}^{t,\star}\tilde{\mathbf{A}}^{t+1} = (\mathbf{A}^{t+1})^{\intercal}\mathbf{A}^{t+1}$. Let $\mathbf{A}^{t,\star}$ denote the low-rank matrix obtained by applying the Procrustes alignment. We first bound $T_1$ as follows:

$$T_1$$
$$= -\eta\mathbb{E}\left[\langle\nabla f(\mathbf{W}^t), \mathbf{L}\left((\mathbf{A}^t)^{\intercal}\bar{\mathbf{G}}_{\mathbf{A}}^t + (\bar{\mathbf{G}}_{\mathbf{A}}^t)^{\intercal}\mathbf{A}^t\right)\mathbf{R}\rangle_F\right] + \eta^2\mathbb{E}\left[\langle\nabla f(\mathbf{W}^t), \mathbf{L}(\bar{\mathbf{G}}_{\mathbf{A}}^t)^{\intercal}\bar{\mathbf{G}}_{\mathbf{A}}^t\mathbf{R}\rangle_F\right]$$
$$= \underbrace{\eta\mathbb{E}\left[\langle\nabla f(\mathbf{W}^t), \mathbf{L}\left((\mathbf{A}^{t,\star}-\mathbf{A}^t)^{\intercal}\bar{\mathbf{G}}_{\mathbf{A}}^t + (\bar{\mathbf{G}}_{\mathbf{A}}^t)^{\intercal}(\mathbf{A}^{t,\star}-\mathbf{A}^t)\right)\mathbf{R}\rangle_F\right]}_{T_3}$$
$$\quad - \eta\mathbb{E}\left[\langle\nabla f(\mathbf{W}^t), \mathbf{L}\left((\mathbf{A}^{t,\star})^{\intercal}\bar{\mathbf{G}}_{\mathbf{A}}^t + (\bar{\mathbf{G}}_{\mathbf{A}}^t)^{\intercal}\mathbf{A}^t\right)\mathbf{R}\rangle_F\right]$$
$$\quad + \eta^2\mathbb{E}\left[\langle\nabla f(\mathbf{W}^t), \mathbf{L}(\bar{\mathbf{G}}_{\mathbf{A}}^t)^{\intercal}\bar{\mathbf{G}}_{\mathbf{A}}^t\mathbf{R}\rangle_F\right]$$
$$\overset{(a)}{=} T_3 - \eta\mathbb{E}\left[\langle\mathbf{H}^t, \left((\mathbf{A}^t)^{\intercal}\bar{\mathbf{G}}_{\mathbf{A}}^t + (\bar{\mathbf{G}}_{\mathbf{A}}^t)^{\intercal}\mathbf{A}^t\right)\rangle_F\right] + \eta^2\mathbb{E}\left[\langle\mathbf{H}^t, (\bar{\mathbf{G}}_{\mathbf{A}}^t)^{\intercal}\bar{\mathbf{G}}_{\mathbf{A}}^t\rangle_F\right]$$

$$= T_3 - \eta\mathbb{E}\left[\left\langle \mathbf{H}^t, (\mathbf{A}^t)^\intercal \mathbf{A}^t\left(\mathbf{H}^t + (\mathbf{H}^t)^\intercal\right) + \left(\mathbf{H}^t + (\mathbf{H}^t)^\intercal\right)(\mathbf{A}^t)^\intercal \mathbf{A}^t\right\rangle_F\right]$$
$$+ \eta^2\mathbb{E}\left[\left\langle \mathbf{H}^t, (\bar{\mathbf{G}}_{\mathbf{A}}^t)^\intercal \bar{\mathbf{G}}_{\mathbf{A}}^t\right\rangle_F\right], \tag{31}$$

where equality (a) is obtained due to the fact that for an arbitrary matrix $\mathbf{X}$, we have $\langle \nabla f(\mathbf{W}^t), \mathbf{LXR}\rangle_F = \langle \mathbf{H}^t, \mathbf{X}\rangle_F$. $T_3$ can be bounded as follows:

$$T_3 \overset{(a)}{\leq} \frac{\beta\eta}{2}\left\|\nabla f(\mathbf{W}^t)\right\|_F^2 + \frac{\eta}{2\beta}\mathbb{E}\left[\left\|\mathbf{L}\left((\mathbf{A}^{t,\star} - \mathbf{A}^t)^\intercal \bar{\mathbf{G}}_{\mathbf{A}}^t + (\bar{\mathbf{G}}_{\mathbf{A}}^t)^\intercal(\mathbf{A}^{t,\star} - \mathbf{A}^t)\right)\mathbf{R}\right\|_F^2\right]$$

$$\overset{(b)}{\leq} \frac{\beta\eta}{2}\left\|\nabla f(\mathbf{W}^t)\right\|_F^2 + \frac{2\eta}{\beta}\mathbb{E}\left[\left\|(\mathbf{A}^{t,\star} - \mathbf{A}^t)^\intercal \bar{\mathbf{G}}_{\mathbf{A}}^t\right\|_F^2\right]$$

$$\overset{(c)}{\leq} \frac{\beta\eta}{2}\left\|\nabla f(\mathbf{W}^t)\right\|_F^2 + \frac{2\eta}{\beta}\left\|\mathbf{A}^{t,\star} - \mathbf{A}^t\right\|_F^2\mathbb{E}\left[\left\|\bar{\mathbf{G}}_{\mathbf{A}}^t\right\|_F^2\right]$$

$$\overset{(d)}{\leq} \frac{\beta\eta}{2}\left\|\nabla f(\mathbf{W}^t)\right\|_F^2 + \frac{2\eta}{\beta N^2}\left\|\mathbf{A}^{t,\star} - \mathbf{A}^t\right\|_F^2\sum_{n\in\mathcal{N}}\mathbb{E}\left[\left\|\nabla_{\mathbf{A}}F_n(\mathbf{W}^t;\boldsymbol{\xi}_n)\right\|_F^2\right]$$

$$\overset{(e)}{\leq} \frac{\beta\eta}{2}\left\|\nabla f(\mathbf{W}^t)\right\|_F^2 + \frac{2\eta\psi}{\beta N}\left\|\mathbf{A}^{t,\star} - \mathbf{A}^t\right\|_F^2$$

$$= \frac{\beta\eta}{2}\left\|\nabla f(\mathbf{W}^t)\right\|_F^2 + \frac{2\eta\psi}{\beta N}\left\|(\mathbf{S}^{t,\star} - \mathbf{S}^t)\tilde{\mathbf{A}}^t\right\|_F^2$$

$$\overset{(f)}{\leq} \frac{\beta\eta}{2}\left\|\nabla f(\mathbf{W}^t)\right\|_F^2 + \frac{2\eta\psi}{\beta N}\left\|\mathbf{S}^{t,\star} - \mathbf{S}^t\right\|_F^2\left\|\tilde{\mathbf{A}}^t\right\|_F^2$$

$$\overset{(g)}{\leq} \frac{\beta\eta}{2}\left\|\nabla f(\mathbf{W}^t)\right\|_F^2 + \frac{2\eta\psi\tilde{C}_{\mathbf{A}}^2\Delta_{\text{proc}}^{t+1}}{\beta N\sigma_{\min}\left(\tilde{\mathbf{A}}^{t+1}(\mathbf{A}^t)^\intercal\right)}, \tag{32}$$

where inequality (a) is obtained due to the fact that $2\langle \mathbf{X}, \mathbf{Y}\rangle_F \leq \beta\|\mathbf{X}\|_F^2 + \frac{1}{\beta}\|\mathbf{Y}\|_F^2$ holds for any $\beta > 0$. Inequalities (b) and (d) result from Jensen's inequality. Inequalities (c) and (f) are obtained by using the fact $\|\mathbf{AB}\|_F \leq \|\mathbf{A}\|_F\|\mathbf{B}\|_F$. Inequality (e) results from Assumption 2. Inequality (g) results from Assumption 3 and Lemma 2. Then, we further bound the second term of $T_1$ as

$$-\eta\mathbb{E}\left[\left\langle \mathbf{H}^t, (\mathbf{A}^t)^\intercal \mathbf{A}^t\left(\mathbf{H}^t + (\mathbf{H}^t)^\intercal\right) + \left(\mathbf{H}^t + (\mathbf{H}^t)^\intercal\right)(\mathbf{A}^t)^\intercal \mathbf{A}^t\right\rangle_F\right]$$
$$\overset{(a)}{\leq} -4\eta\lambda_{\min}\left((\mathbf{A}^t)^\intercal \mathbf{A}^t\right)\left\|(\mathbf{L})^\intercal\nabla f(\mathbf{W}^t)(\mathbf{R})^\intercal\right\|_F^2, \tag{33}$$

where inequality (a) results from Lemma 1. Then, we bound $\eta^2\mathbb{E}\left[\left\langle \mathbf{H}^t, (\bar{\mathbf{G}}_{\mathbf{A}}^t)^\intercal \bar{\mathbf{G}}_{\mathbf{A}}^t\right\rangle_F\right]$ as follows:

$$\eta^2\mathbb{E}\left[\left\langle \mathbf{H}^t, (\bar{\mathbf{G}}_{\mathbf{A}}^t)^\intercal \bar{\mathbf{G}}_{\mathbf{A}}^t\right\rangle_F\right] \overset{(a)}{\leq} \eta^2\left\|\mathbf{H}^t\right\|_F\mathbb{E}\left[\left\|(\bar{\mathbf{G}}_{\mathbf{A}}^t)^\intercal \bar{\mathbf{G}}_{\mathbf{A}}^t\right\|_F\right]$$

$$\overset{(b)}{\leq} \eta^2\left\|\mathbf{H}^t\right\|_F\mathbb{E}\left[\left\|\bar{\mathbf{G}}_{\mathbf{A}}^t\right\|_F^2\right]$$

$$\overset{(c)}{\leq} \eta^2\left\|\mathbf{H}^t\right\|_F\psi$$

$$\overset{(d)}{\leq} \frac{\eta^2}{2}\left\|\mathbf{H}^t\right\|_F^2 + \frac{\eta^2}{2}\psi^2$$

$$\leq \frac{\eta^2}{2}\left\|(\mathbf{L})^\intercal\nabla f(\mathbf{W}^t)(\mathbf{R})^\intercal\right\|_F^2 + \frac{\eta^2\psi^2}{2}, \tag{34}$$

where inequality (a) follows from Cauchy-Schwarz inequality. Inequality (b) results from the fact that $\|(\bar{\mathbf{G}}_{\mathbf{A}}^t)^\intercal \bar{\mathbf{G}}_{\mathbf{A}}^t\|_F \leq \|\bar{\mathbf{G}}_{\mathbf{A}}^t\|_F^2$. Inequality (c) is obtained by using Assumption 2. Inequality (d) follows from Young's inequality.

By combining inequalities (31), (32), (33), and (34), we have

$$T_1 \leq \frac{2\eta\psi\tilde{C}_{\mathbf{A}}^2\Delta_{\text{proc}}^{t+1}}{\beta N\sigma_{\min}\left(\tilde{\mathbf{A}}^{t+1}(\mathbf{A}^t)^\intercal\right)} + \left(-4\eta\lambda_{\min}\left((\mathbf{A}^t)^\intercal \mathbf{A}^t\right) + \frac{\eta^2}{2} + \frac{\beta\eta}{2}\right)\left\|\nabla f(\mathbf{W}^t)\right\|_F^2 + \frac{\eta^2}{2}\psi^2.$$
$$\tag{35}$$

Now, we bound $T_2$. In particular, it satisfies

$$
\begin{aligned}
T_2 &= \mathbb{E}\left[\left\|\mathbf{L}\left(\eta^2(\bar{\mathbf{G}}_{\mathbf{A}}^t)^{\intercal}\bar{\mathbf{G}}_{\mathbf{A}}^t - \eta\left((\mathbf{A}^t)^{\intercal}\bar{\mathbf{G}}_{\mathbf{A}}^t + (\bar{\mathbf{G}}_{\mathbf{A}}^t)^{\intercal}\mathbf{A}^t\right)\right)\mathbf{R}\right\|_F^2\right] \\
&\stackrel{(a)}{\leq} 3\eta^4\mathbb{E}\left[\left\|\mathbf{L}\eta^2(\bar{\mathbf{G}}_{\mathbf{A}}^t)^{\intercal}\bar{\mathbf{G}}_{\mathbf{A}}^t\mathbf{R}\right\|_F^2\right] + 6\eta^2\mathbb{E}\left[\left\|\mathbf{L}\eta(\mathbf{A}^t)^{\intercal}\bar{\mathbf{G}}_{\mathbf{A}}^t\mathbf{R}\right\|_F^2\right] \\
&\stackrel{(b)}{\leq} 3\eta^4\mathbb{E}\left[\left\|(\bar{\mathbf{G}}_{\mathbf{A}}^t)^{\intercal}\bar{\mathbf{G}}_{\mathbf{A}}^t\right\|_F^2\right] + 6\eta^2\mathbb{E}\left[\left\|(\mathbf{A}^t)^{\intercal}\bar{\mathbf{G}}_{\mathbf{A}}^t\right\|_F^2\right] \\
&\stackrel{(c)}{\leq} 3\eta^4\mathbb{E}\left[\left\|\bar{\mathbf{G}}_{\mathbf{A}}^t\right\|_F^4\right] + 6\eta^2\mathbb{E}\left[\left\|\mathbf{A}^t\right\|_F^2\right]\mathbb{E}\left[\left\|\bar{\mathbf{G}}_{\mathbf{A}}^t\right\|_F^2\right] \\
&\stackrel{(d)}{\leq} 3\eta^2\psi\left(\eta^2\psi + 2C_{\mathbf{A}}^2\right).
\end{aligned}
\tag{36}
$$

where inequality (a) is obtained by using Jensen's inequality. Inequality (b) is obtained since $\mathbf{L}$ and $\mathbf{R}$ have orthogonal columns. Inequality (c) results from the fact $\|\mathbf{AB}\|_F \leq \|\mathbf{A}\|_F\|\mathbf{B}\|_F$. Inequality (d) is obtained by using Assumption 2 and Assumption 3.

Then, we combine inequalities (29), (35), and (36), we have

$$
\begin{aligned}
f(\mathbf{W}^{t+1}) &\leq f(\mathbf{W}^t) + \left(\frac{\eta^2}{2} + \frac{\beta\eta}{2} - 4\eta\lambda_{\min}\left((\mathbf{A}^t)^{\intercal}\mathbf{A}^t\right)\right)\left\|(\nabla f(\mathbf{W}^t)\right\|_F^2 \\
&+ \frac{\eta^2\psi^2}{2} + \frac{3L\eta^2\psi\left(\eta^2\psi + 2C_{\mathbf{A}}^2\right)}{2} + \frac{2\eta\psi\tilde{C}_{\mathbf{A}}^2\Delta_{\text{proc}}^{t+1}}{\beta N\sigma_{\min}\left(\tilde{\mathbf{A}}^{t+1}(\mathbf{A}^t)^{\intercal}\right)}.
\end{aligned}
\tag{37}
$$

We denote $\Omega = 4\eta\min_{t\in\mathcal{T}}\{\lambda_{\min}\left((\mathbf{A}^t)^{\intercal}\mathbf{A}^t\right)\} - \frac{\eta^2}{2} - \frac{\beta\eta}{2}$. We choose $\beta = 4\min_{t\in\mathcal{T}}\{\lambda_{\min}\left((\mathbf{A}^t)^{\intercal}\mathbf{A}^t\right)\}$. Then, we have $\Omega = 2\eta\min_{t\in\mathcal{T}}\{\lambda_{\min}\left((\mathbf{A}^t)^{\intercal}\mathbf{A}^t\right)\} - \frac{\eta^2}{2}$. By rearranging inequality (37) and summing over all $T$ fine-tuning rounds, we have

$$
\begin{aligned}
\Omega\sum_{t\in\mathcal{T}}\left\|\nabla f(\mathbf{W}^t)\right\|_F^2 &\leq \sum_{t\in\mathcal{T}}\left(f(\mathbf{W}^t) - f(\mathbf{W}^{t+1})\right) + T\left(\frac{\eta^2\psi^2}{2} + \frac{3L\eta^2\psi\left(\eta^2\psi + 2C_{\mathbf{A}}^2\right)}{2}\right) \\
&+ \sum_{t\in\mathcal{T}}\frac{\eta\psi\tilde{C}_{\mathbf{A}}^2\Delta_{\text{proc}}^{t+1}}{2N\min_{t\in\mathcal{T}}\{\lambda_{\min}\left((\mathbf{A}^t)^{\intercal}\mathbf{A}^t\right)\}\sigma_{\min}\left(\tilde{\mathbf{A}}^{t+1}(\mathbf{A}^t)^{\intercal}\right)} \\
&\stackrel{(a)}{\leq} f(\mathbf{W}^1) - f(\mathbf{W}^{\star}) + T\left(\frac{\eta^2\psi^2}{2} + \frac{3L\eta^2\psi\left(\eta^2\psi + 2C_{\mathbf{A}}^2\right)}{2}\right) \\
&+ \frac{\eta\psi\tilde{C}_{\mathbf{A}}^2\sum_{t\in\mathcal{T}}\frac{\Delta_{\text{proc}}^{t+1}}{\sigma_{\min}(\tilde{\mathbf{A}}^{t+1}(\mathbf{A}^t)^{\intercal})}}{2N\min_{t\in\mathcal{T}}\{\lambda_{\min}\left((\mathbf{A}^t)^{\intercal}\mathbf{A}^t\right)\}},
\end{aligned}
\tag{38}
$$

where inequality (a) is obtained due to the fact that $f(\mathbf{W}^{\star}) \leq f(\mathbf{W}^{T+1})$. If $\eta < 4\min_{t\in\mathcal{T}}\{\lambda_{\min}\left((\mathbf{A}^t)^{\intercal}\mathbf{A}^t\right)\}$, we have $\Omega > 0$. Then, we multiply $\frac{1}{T\Omega}$ on both sides of inequality (38) and obtain

$$
\begin{aligned}
\frac{1}{T}\sum_{t\in\mathcal{T}}\left\|\nabla f(\mathbf{W}^t)\right\|_F^2 &\leq \frac{f(\mathbf{W}^1) - f(\mathbf{W}^{\star})}{T\Omega} + \frac{\eta^2\psi^2}{2\Omega} + \frac{3L\eta^2\psi\left(\eta^2\psi + 2C_{\mathbf{A}}^2\right)}{2\Omega} \\
&+ \frac{\eta\psi\tilde{C}_{\mathbf{A}}^2\sum_{t\in\mathcal{T}}\frac{\Delta_{\text{proc}}^{t+1}}{\sigma_{\min}(\tilde{\mathbf{A}}^{t+1}(\mathbf{A}^t)^{\intercal})}}{2NT\Omega\min_{t\in\mathcal{T}}\{\lambda_{\min}\left((\mathbf{A}^t)^{\intercal}\mathbf{A}^t\right)\}}.
\end{aligned}
\tag{39}
$$

This completes the proof of Theorem 2.

### A.6 HYPERPARAMETER SETTING

In our experiments, the batch size is set to be 4 by searching over a range of $\{2, 4, 8, 16\}$. Each local training epoch is set to be 1. We select the optimal learning rate from the range $\{5e-4, 1e-$

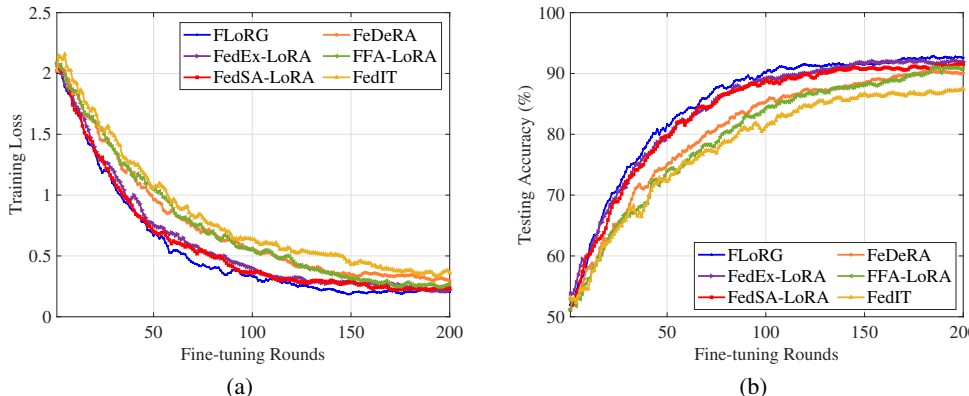

Figure 2: Comparison of the (a) training loss and (b) testing accuracy versus the number of fine-tuning rounds on RoBERTa-large and QNLI.

Table 8: Comparison of the per-round client-side computation overhead (in FLOPs), per-round server-side computation overhead (in FLOPs), and memory usage (in MB). Symbol "−" means that the overhead is negligible.

| Metrics | FLoRG | FedIT | FeDeRA | FFA-LoRA | FedSA-LoRA | FedEx-LoRA |
|---|---|---|---|---|---|---|
| Client-side FLOPs | $1.18 \times 10^{12}$ | $1.24 \times 10^{12}$ | $1.24 \times 10^{12}$ | $\mathbf{8.25 \times 10^{11}}$ | $1.24 \times 10^{12}$ | $1.18 \times 10^{12}$ |
| Server-side FLOPs | $8.50 \times 10^{10}$ | "−" | $2.22 \times 10^{11}$ | "−" | "−" | $2.09 \times 10^{9}$ |
| Memory Usage | 2815 | 2117 | 2117 | **2110** | 2117 | 3117 |

$4, 5e - 5, 1e - 5\}$. The maximum sequence length is set to be 128, following the common practice in fine-tuning. We choose AdamW as the optimizer. We set the scaling factor of the LoRA module to be 16 for all algorithms.

## A.7 ADDITIONAL EXPERIMENTAL RESULTS

**Convergence curve:** Fig. 2 reports the training loss and testing accuracy versus the number of fine-tuning rounds on RoBERTa-large with the QNLI dataset. We observe that FLoRG consistently converges faster than the five baselines: it achieves a lower training loss throughout the training process and reaches a higher testing accuracy with fewer rounds. This indicates that the proposed Gram-matrix aggregation with the Procrustes alignment stabilizes the optimization trajectory across fine-tuning rounds, leading to both improved convergence speed and better final generalization performance.

**Computation overhead and memory usage:** Table 8 compares the per-round computation overhead (client-side and server-side FLOPs) and the memory usage. On the client side, FLoRG incurs $1.18 \times 10^{12}$ FLOPs per round, which is comparable to FedIT/FeDeRA/FedSA-LoRA ($1.24 \times 10^{12}$) and matches FedEx-LoRA ($1.18 \times 10^{12}$), while FFA-LoRA achieves a lower client-side FLOPs ($8.25 \times 10^{11}$) due to freezing one low-rank factor. On the server side, FLoRG introduces a non-negligible overhead ($8.50 \times 10^{10}$ FLOPs) because it performs eigendecomposition and Procrustes alignment after Gram aggregation; nevertheless, its server-side overhead is substantially lower than FeDeRA ($2.22 \times 10^{11}$ FLOPs), which relies on SVD-based decomposition. In terms of memory usage, FLoRG uses 2815 MB, which is higher than most baselines that only maintain two low-rank matrices (around 2110–2117 MB), but remains comparable in scale and is lower than FedEx-LoRA (3117 MB).

**Question answering performance on SQuAD v1.1:** Table 9 reports the exact match (EM) and F1 scores on SQuAD v1.1 under three base models. FLoRG achieves the best performance across all models and datasets. For OPT-125M, FLoRG improves EM/F1 to 68.52/78.23, outperforming the strongest baseline (e.g., FedSA-LoRA) by 1.38 EM and 0.92 F1. For RoBERTa-large, FLoRG

Table 9: Comparison of the exact match (EM) and F1 scores on SQuAD v1.1.

| Base Model | Metric | FLoRG | FedIT | FeDeRA | FFA-LoRA | FedSA-LoRA | FedEx-LoRA |
|---|---|---|---|---|---|---|---|
| OPT-125M | EM | **68.52** | 62.31 | 64.78 | 63.19 | 67.14 | 66.85 |
| | F1 | **78.23** | 72.08 | 74.52 | 73.04 | 77.31 | 76.18 |
| RoBERTa-large | EM | **86.34** | 81.53 | 83.12 | 81.69 | 84.47 | 86.02 |
| | F1 | **91.15** | 88.09 | 90.03 | 88.68 | 90.23 | 90.84 |
| Llama-3.2-3B | EM | **89.83** | 84.21 | 86.58 | 85.37 | 87.12 | 88.25 |
| | F1 | **92.74** | 89.53 | 91.08 | 90.31 | 91.35 | 91.82 |

Table 10: Additional comparison of the testing accuracy across different baseline schemes.

| Base Model | Dataset | FLoRG | FedIT | FeDeRA | FFA-LoRA | FedSA-LoRA |
|---|---|---|---|---|---|---|
| OPT-125M | MRPC | **88.30** | 82.09 | 85.70 | 84.78 | 86.51 |
| | QQP | **89.72** | 84.18 | 87.09 | 86.30 | 87.92 |
| RoBERTa-large | MRPC | **90.80** | 85.16 | 88.30 | 87.50 | 89.21 |
| | QQP | **91.52** | 87.89 | 89.30 | 88.51 | 90.01 |

reaches 86.34 EM and 91.15 F1, slightly exceeding the best competing method (FedEx-LoRA) by 0.32 EM and 0.31 F1. For Llama-3.2-3B, FLoRG obtains 89.83 EM and 92.74 F1, improving over the strongest baseline (FedEx-LoRA) by 1.58 EM and 0.92 F1. These results demonstrate that FLoRG generalizes beyond GLUE classification tasks and is also effective for extractive question and answering.

**Additional GLUE results:** To complement Table 1 in the main content, Table 10 further reports the testing accuracy on MRPC and QQP datasets. FLoRG consistently outperforms the baselines on both OPT-125M and RoBERTa-large. In particular, on OPT-125M, FLoRG improves MRPC and QQP to 88.30 and 89.72, exceeding FedSA-LoRA by 1.79 and 1.80, respectively. On RoBERTa-large, FLoRG achieves 90.80 (MRPC) and 91.52 (QQP), exceeding FedSA-LoRA by 1.59 and 1.51, respectively. This reinforces the effectiveness of FLoRG across different GLUE tasks.

**Additional rank study:** Table 11 provides more detailed results under different ranks ($r \in \{2, 4, 8\}$) on MRPC, QQP, MNLI, and QNLI (RoBERTa-large). FLoRG remains the top-performing method under all rank settings. Moreover, increasing $r$ generally improves the testing accuracy for all methods, while the performance gap between FLoRG and the baselines persists, indicating that FLoRG is robust to rank choices and can better exploit higher-rank adaptation capacity when available.

**Robustness to data heterogeneity:** Table 12 reports additional results under different degrees of non-IIDness controlled by $\rho$. Under highly heterogeneous settings (e.g., $\rho = 0.1$), FLoRG consistently outperforms the baseline schemes on MRPC/QQP/MNLI/QNLI. When the data distribution becomes less heterogeneous ($\rho = 0.5$), FLoRG still maintains consistent improvements over competing methods. In the near-IID case ($\rho = 1$), FLoRG remains competitive and achieves the best results on MRPC/QQP/MNLI, while FedSA-LoRA slightly surpasses FLoRG on QNLI. Overall, these results validate that FLoRG is robust across a wide range of client data heterogeneity levels, with particularly strong advantages under more heterogeneous federated settings.

## A.8 THE USE OF LARGE LANGUAGE MODELS (LLMS)

We used large language models (e.g., ChatGPT) for the general purpose of writing assistants to revise and polish the manuscript's prose: improving grammar, wording, clarity, and fixing minor LaTeX/formatting issues. The models were not used for research ideation, designing algorithms/experiments, deriving proofs, selecting citations, or writing substantive technical content.

Table 11: Additional comparison of the testing accuracy under different ranks.

| Rank | Dataset | FLoRG | FedIT | FeDeRA | FFA-LoRA | FedSA-LoRA |
|------|---------|-------|-------|--------|----------|------------|
| $r = 2$ | MRPC | **85.91** | 79.66 | 83.17 | 82.32 | 83.00 |
| | QQP | **87.62** | 81.88 | 84.90 | 84.09 | 85.65 |
| | MNLI | **86.14** | 80.70 | 83.44 | 82.64 | 84.22 |
| | QNLI | **89.70** | 86.67 | 87.22 | 86.40 | 88.10 |
| $r = 4$ | MRPC | **90.80** | 85.16 | 88.30 | 87.50 | 89.21 |
| | QQP | **91.52** | 87.89 | 89.30 | 88.51 | 90.01 |
| | MNLI | **91.39** | 84.76 | 88.20 | 89.12 | 90.90 |
| | QNLI | **92.44** | 87.63 | 89.91 | 90.82 | 91.69 |
| $r = 8$ | MRPC | **91.70** | 85.55 | 87.12 | 88.20 | 89.91 |
| | QQP | **91.80** | 86.39 | 89.26 | 88.45 | 90.10 |
| | MNLI | **91.70** | 86.40 | 89.29 | 90.44 | 90.41 |
| | QNLI | **92.50** | 87.77 | 90.20 | 90.41 | 91.13 |

Table 12: Additional comparison of the testing accuracy under different degrees of data heterogeneity.

| Non-IIDness | Dataset | FLoRG | FedIT | FeDeRA | FFA-LoRA | FedSA-LoRA |
|-------------|---------|-------|-------|--------|----------|------------|
| $\rho = 0.1$ | MRPC | **85.06** | 78.19 | 81.91 | 81.00 | 83.01 |
| | QQP | **86.95** | 78.80 | 84.15 | 83.21 | 85.00 |
| | MNLI | **81.35** | 75.44 | 78.42 | 77.57 | 79.36 |
| | QNLI | **88.88** | 83.31 | 86.20 | 85.33 | 87.19 |
| $\rho = 0.5$ | MRPC | **90.80** | 85.16 | 88.30 | 87.50 | 89.21 |
| | QQP | **91.52** | 87.89 | 89.30 | 88.51 | 90.01 |
| | MNLI | **91.39** | 84.76 | 88.20 | 89.12 | 90.90 |
| | QNLI | **92.44** | 87.63 | 89.91 | 90.82 | 91.69 |
| $\rho = 1$ | MRPC | **90.76** | 85.77 | 89.26 | 88.91 | 88.20 |
| | QQP | **91.88** | 88.61 | 89.93 | 89.37 | 90.39 |
| | MNLI | **91.75** | 86.47 | 90.05 | 91.70 | 91.33 |
| | QNLI | 92.17 | 88.03 | 90.52 | 90.79 | **92.72** |

