# OpenReview forum: "FLoRG: Federated Fine-tuning with Low-rank Gram Matrices and Procrustes Alignment"
_ICLR.cc/2026/Conference — ICLR 2026 Poster_

### Official Review · Reviewer_kKoy · 2025-10-30

**Soundness:** 3
**Presentation:** 3
**Contribution:** 3
**Rating:** 4
**Confidence:** 3

**Summary:**

The paper proposes FLoRG, a novel federated fine-tuning framework for large language models (LLMs) that improves upon the conventional LoRA-based approach. Traditional federated LoRA uses two separate low-rank matrices, which can lead to aggregation errors and decomposition drift during global model updates. To address these issues, FLoRG employs a single low-rank matrix and aggregates its Gram matrix to eliminate aggregation errors and reduce communication costs. Additionally, the authors introduce a Procrustes alignment mechanism to align decomposed matrices between rounds, thereby mitigating decomposition drift. Theoretical analysis demonstrates that this design achieves a tighter convergence bound. Extensive experiments on multiple LLM fine-tuning benchmarks show that FLoRG achieves higher downstream task accuracy and reduces communication overhead by up to 82% compared to state-of-the-art baselines.

**Strengths:**

1. The authors provide a theoretical convergence analysis and rigorously show that the proposed Procrustes alignment leads to a tighter convergence bound, which enhances the credibility of the method.
2. Experimental results on multiple LLM fine-tuning benchmarks demonstrate consistent improvements over several state-of-the-art baselines in both accuracy and communication efficiency.
3. The paper is well written and easy to follow.

**Weaknesses:**

1. The experiments are limited to natural language understanding tasks with relatively small models. It would strengthen the paper to include evaluations on larger models or additional task types to demonstrate broader applicability.
2. Updating only the low-rank matrix  $A$ may limit the model’s representation capability, potentially constraining its ability to adapt to more complex tasks.
3. The paper lacks an analysis of efficiency in terms of computational cost, memory usage, or communication overhead.

**Questions:**

1. How does the proposed method perform on more general tasks such as question answering and dialogue?
2. The performance of the proposed method when applied to more popular and larger language models, such as LLaMA?
3. The local training stage of the proposed method is interesting. It seems applicable to centralized learning as well. How does it perform under centralized learning? Compared with centralized learning, what advantages does it provide in the FL setting?
4. It is unclear whether updating only module A is sufficient to learn good representations of the client data. Could the authors clarify or provide evidence?
5. Could the authors provide details on how the dataset was used?

If the authors can adequately address my concerns, I am willing to increase my rating.

---

> ### Author Response · Authors · 2025-11-21
>
> **Weakness 1:** The experiments are limited to natural language understanding tasks with relatively small models. It would strengthen the paper to include evaluations on larger models or additional task types to demonstrate broader applicability.
>
> **Author's Response:** Thank you for pointing this out.
> First, in Section 4 of the revised manuscript, we have included LLama-3.2-3B for our experiments.
> The results show that our proposed FLoRG outperforms the baseline schemes
> under those four datasets in most cases.
>
> **Table 1:**
>
> | Base Model      | Dataset | FLoRG     | FedIT  | FeDeRA | FFA-LoRA | FedSA-LoRA | FedEx-LoRA |
> |-----------------|:--------|:---------:|:------:|:------:|:--------:|:----------:|:----------:|
> | OPT-125M        | MNLI    | **87.35** | 79.42  | 81.15  | 83.54    | 84.61      | 85.83      |
> | OPT-125M        | QNLI    | 89.52     | 84.18  | 86.71  | 87.93    | 88.69      | **89.88**  |
> | OPT-125M        | WNLI    | **65.28** | 58.45  | 59.34  | 62.61    | 62.83      | 64.15      |
> | OPT-125M        | RTE     | **68.92** | 61.08  | 64.51  | 66.02    | 67.39      | 68.27      |
> | RoBERTa-large   | MNLI    | **91.27** | 84.91  | 88.06  | 89.28    | 90.75      | 90.96      |
> | RoBERTa-large   | QNLI    | **92.58** | 87.49  | 90.07  | 90.96    | 91.54      | 92.13      |
> | RoBERTa-large   | WNLI    | **66.48** | 59.34  | 61.83  | 63.72    | 63.47      | 66.11      |
> | RoBERTa-large   | RTE     | **71.26** | 64.25  | 67.12  | 68.49    | 69.93      | 70.98      |
> | LLaMa-3.2-3B    | MNLI    | **93.15** | 87.24  | 89.83  | 91.05    | 92.38      | 92.74      |
> | LLaMa-3.2-3B    | QNLI    | 93.12     | 89.17  | 91.45  | 92.53    | **93.27**  | 93.05      |
> | LLaMa-3.2-3B    | WNLI    | **68.73** | 61.52  | 64.19  | 65.97    | 66.81      | 67.89      |
> | LLaMa-3.2-3B    | RTE     | **73.84** | 67.08  | 69.75  | 71.33    | 72.56      | 73.15      |
>
> Second, in Appendix A.7 of the revised manuscript, we have included SQuAD v1.1 as the question-answering dataset task for our experiment. The results in Table 9 show that our proposed FLoRG outperforms the baseline schemes
> under different base models.
>
> **Table 9:**
>
> | Base Model      | Metric | FLoRG   | FedIT  | FeDeRA | FFA-LoRA | FedSA-LoRA | FedEx-LoRA |
> |-----------------|--------|---------|--------|--------|----------|------------|------------|
> | OPT-125M        | EM     | **68.52** | 62.31  | 64.78  | 63.19    | 67.14      | 66.85      |
> | OPT-125M        | F1     | **78.23** | 72.08  | 74.52  | 73.04    | 77.31      | 76.18      |
> | RoBERTa-large   | EM     | **86.34** | 81.53  | 83.12  | 81.69    | 84.47      | 86.02      |
> | RoBERTa-large   | F1     | **91.15** | 88.09  | 90.03  | 88.68    | 90.23      | 90.84      |
> | LLaMA-3.2-3B    | EM     | **89.83** | 84.21  | 86.58  | 85.37    | 87.12      | 88.25      |
> | LLaMA-3.2-3B    | F1     | **92.74** | 89.53  | 91.08  | 90.31    | 91.35      | 91.82      |
>
> [1] Yuxuan Yan, Qianqian Yang, Shunpu Tang, and Zhiguo Shi. FeDeRA: Efficient fine-tuning of language models in federated learning leveraging weight decomposition. In _Proc. Advances Neural
> Info. Process. Syst. (NeurIPS)_, Vancouver, Canada, Dec. 2024.
>
> [2] Raghav Singhal, Kaustubh Ponkshe, and Praneeth Vepakomma. FedEx-LoRA: Exact aggregation
> for federated and efficient fine-tuning of foundation models. In _Proc. Annu. Meeting Assoc. Com-
> put. Linguistics_, Vienna, Austria, Jul. 2025.
>
> **Weakness 2:** Updating only the low-rank matrix A may limit the model’s representation capability, potentially constraining its ability to adapt to more complex tasks.
>
> **Author's Response:** We appreciate the reviewer's comment.
> In FLoRG, each layer’s update is parameterized by a single low-rank matrix $\mathbf{A}$ per client, which is then mapped through shared semi-orthogonal bases to produce a full update.
> This preserves a rich set of directions in the original parameter space while allowing us to (i) reduce the number of trainable and transmitted model parameters by only updating $\mathbf{A}$, and (ii) avoid the scale imbalance and optimizer-sensitivity issues known for the two-matrix $\mathbf{BA}$ parameterization in conventional LoRA approaches.

---

> ### Author Response · Authors · 2025-11-21
>
> **Author's Response Continued:** In terms of additional tasks, please refer to the response to weakness 1.
> The results in Table 9 of the revised manuscript validate that updating a single low-rank matrix does not limit the model’s representation capability.
> In addition, it outperforms baseline schemes which update two low-rank matrices in most cases.
>
> **Weakness 3:** The paper lacks an analysis of efficiency in terms of computational cost, memory usage, or communication overhead.
>
> **Author's Response:** Thank you for pointing this out.  In Table 3 of thr revised manuscript, we present the comparison of the communication overhead.
>
> **Table 3:**
>
> | Base Model     | Target Acc. | FLoRG            | FedIT           | FeDeRA          | FFA-LoRA        | FedSA-LoRA      | FedEx-LoRA        |
> |----------------|-------------|------------------|-----------------|-----------------|-----------------|-----------------|-------------------|
> | OPT-125M       | 80.00       | **8.2 × 10^6**   | 3.78 × 10^7     | 2.46 × 10^7     | 1.59 × 10^7     | 2.10 × 10^7     | 1.25 × 10^10      |
> | OPT-125M       | 85.00       | **1.07 × 10^7**  | -               | 4.2 × 10^7      | 2.75 × 10^7     | 3.61 × 10^7     | 1.77 × 10^10      |
> | RoBERTa-large  | 80.00       | **8.7 × 10^6**   | 4.68 × 10^7     | 3.02 × 10^7     | 1.85 × 10^7     | 2.59 × 10^7     | 2.08 × 10^10      |
> | RoBERTa-large  | 85.00       | **1.45 × 10^7**  | 8.12 × 10^7     | 5.17 × 10^7     | 3.35 × 10^7     | 4.42 × 10^7     | 2.96 × 10^10      |
>
> show that to achieve the target test accuracy, our proposed FLoRG uses a much lower total number of transmitted model parameters when compared with the baselines.
> This demonstrates that FLoRG can significantly reduce the communication overhead by up to 2041$\times$ when compared with the baseline schemes.
> Additional results of the computation overhead and memory usage are presented in Table 8 in Appendix A.7 of the revised manuscript.
>
> **Table 8:**
>
> | Metrics              | FLoRG             | FedIT             | FeDeRA            | FFA-LoRA             | FedSA-LoRA         | FedEX-LoRA         |
> |----------------------|-------------------|-------------------|-------------------|----------------------|--------------------|--------------------|
> | Client-side FLOPs    | 1.18 × 10^12      | 1.24 × 10^12      | 1.24 × 10^12      | **8.25 × 10^11**     | 1.24 × 10^12       | 1.18 × 10^12       |
> | Server-side FLOPs    | 8.50 × 10^10      | -                 | 2.22 × 10^11      | -                    | -                  | 2.09 × 10^9        |
> | Memory Usage (MB)    | 2815              | 2117              | 2117              | **2110**             | 2117               | 3107               |
>
> **Question 1:** How does the proposed method perform on more general tasks such as question answering and dialogue?
>
> **Author's Response:** We thank the reviewer for this suggestion.
> Please refer to our response in weakness 1.
> The results in Table 9 show that our proposed FLoRG outperforms the baseline schemes under different base models.
> Architecturally, FLoRG is task-agnostic: it only changes how LoRA modules are parameterized and aggregated across clients, and it can be plugged into any backbone and fine-tuning pipeline where standard LoRA is used (including extractive Q\&A and dialogues).
>
> Again, we acknowledge that including experiments under more task types would further enrich the evaluation. However, due to the limited rebuttal period, it is very challenging to conduct another new line of experiments with additional task types under three backbone models with different scales and six comparison schemes.
> We hope that your insightful comment can motivate more comprehensive studies.
>
> **Question 2:** The performance of the proposed method when applied to more popular and larger language models, such as LLaMA?
>
> **Author's Response:** Thank you for pointing this out. In Table 1 of the revised manuscript (please also refer to Table 1 in the response to Weakness 1), we have included LLama-3.2-3B for our experiments. The results show that our proposed FLoRG outperforms the baseline schemes under those four datasets in most cases.

---

> ### Author Response · Authors · 2025-11-21
>
> **Question 3:** The local training stage of the proposed method is interesting. It seems applicable to centralized learning as well. How does it perform under centralized learning? Compared with centralized learning, what advantages does it provide in the FL setting?
>
> **Author's Response:** We thank the reviewer for the insightful comment.
> The local training stage of FLoRG is applicable to the centralized learning.
> If we disable client-server communication in FLoRG and run the algorithm on a single machine, our update reduces to learning a single low-rank matrix whose Gram matrix is $(\mathbf{A}^{t})^{\intercal}\mathbf{A}^{t}$, similar to recent single-matrix low-rank adapters such as SingLoRA [1].
> For your reference, the key difference of the local training of FLoRG from SingLoRA is that our update naturally handles rectangular parameter matrices due to the shared semi-orthogonal bases $\mathbf{L}$ and $\mathbf{R}$.
>
> The main advantages of FLoRG appear in the FL setting: it provides:
> - a strict reduction in communicated parameters per round since only matrix $\mathbf{A}$ is transmitted,
> - a principled way to aggregate client updates via the global Gram matrix and Procrustes alignment,
> - a way to avoid the scale imbalance and optimizer-sensitivity issues known for the two-matrix $\mathbf{BA}$ parameterization in conventional LoRA approaches.
>
> To sum up, FLoRG is designed primarily for FL settings, but its local training formulation can naturally generalize to centralized learning as well.
>
> [1] David Bensa¨ıd, Noam Rotstein, Roy Velich, Daniel Bensa¨ıd, and Ron Kimmel. SingLoRA: Low
> rank adaptation using a single matrix. _arXiv preprint arXiv:2507.05566_, Jul. 2025.
>
> **Question 4:** It is unclear whether updating only module A is sufficient to learn good representations of the client data. Could the authors clarify or provide evidence?
>
> **Author's Response:** Thank you for your comment. Please refer to the response to weakness 2. In addition, in Table I of the revised manuscript (please also refer to Table 1 in the response to Weakness 1), we compare the performance of FLoRG with the other five baseline schemes, most of which update two low-rank matrices (i.e., both $\mathbf{B}$ and $\mathbf{A}$), whereas FFA-LoRA updates only $\mathbf{B}$. The results show that our proposed FLoRG outperforms the baseline schemes under those four datasets in most cases, underscoring the ability of FLoRG to learn good representations of the client data.
>
> **Question 5:** Could the authors provide details on how the dataset was used?
>
> **Author's Response:** We thank the reviewer for pointing this out.
> As described in Section 4.1, we use GLUE as the benchmark dataset for natural language understanding and consider six standard supervised tasks: MRPC, QQP, MNLI, QNLI, WNLI, and RTE.
> In addition, we use SQuAD v1.1 as the question-answering dataset.
> For the question-answering task with SQuAD v1.1, we consider an iid setting.
> For each dataset within GLUE, we follow the official GLUE train/validation splits and use the corresponding labels.
> In the FL simulation, we evenly allocate the same number of training samples to clients to ensure fairness.
> In particular, for each dataset, we sample client-specific label proportions $\boldsymbol{\pi}_n \sim \mathrm{Dir}(\rho)$, and allocate training samples to client $n$ according to $\boldsymbol{\pi}_n$.
> The scalar parameter $\rho>0$ controls the degree of data heterogeneity: a smaller $\rho$ yields more skewed label distributions, while a larger $\rho$ yields more balanced distributions.
> Each client then trains locally on its own partition, and we report the testing accuracy of the trained model.

---

> ### Author Response · Authors · 2025-11-21
>
> We thank the reviewer for his/her constructive comments.
> We appreciate his/her time for reviewing the paper.
> We hope that we have addressed the comments in a satisfactory manner.

---

### Official Review · Reviewer_fXbx · 2025-10-30

**Soundness:** 3
**Presentation:** 3
**Contribution:** 3
**Rating:** 6
**Confidence:** 3

**Summary:**

FLoRG replaces the usual LoRA two-factor update BA with a single low-rank matrix A and updates the model via a Gram matrix aggregation. Clients locally SGD-update A; the server linearly aggregates n∑An⊤An, then eigendecomposes the aggregated Gram and applies a Procrustes alignment to pick a decomposition closest to the previous A, which stabilizes directions and enforces a target rank. The authors prove a non-convex convergence bound in which Procrustes alignment cancels a “drift” term, and empirically report higher GLUE accuracy vs. FedIT/FeDeRA/FFA-LoRA/FedSA-LoRA with up to 82% fewer transmitted parameters to reach target accuracy.

**Strengths:**

1. Bias-free aggregation with one matrix.
2. Convergence bound tightens when alignment is used; ablations show sizeable accuracy gains from Procrustes; headline comms savings to target accuracy.

**Weaknesses:**

1. The approach relies on semi-orthogonal L,R that never update; performance is sensitive to their initialization.
2. Each round per layer requires eigendecomposition of Q and an SVD for Procrustes; scalability or latency with many layers or clients isn’t benchmarked.

**Questions:**

1. How robust is FLoRG if L,R are learned (slowly) or adapted per layer/round? Can you provide theory/ablation for updating L,R versus keeping them fixed?
2. What are per-round costs of eigendecomposition + Procrustes across all LoRA layers at N>100 clients?

---

> ### Author Response · Authors · 2025-11-21
>
> **Weakness 1:** The approach relies on semi-orthogonal L,R that never update; performance is sensitive to their initialization.
>
> **Author's Response:** We appreciate the reviewer's concern.
> We acknowledge that in FLoRG, the matrices $\mathbf{L}\in\mathbb{R}^{d_{\mathrm{out}}\times k}$ and $\mathbf{R}\in\mathbb{R}^{k\times d_{\mathrm{in}}}$ are shared semi-orthogonal bases which remain fixed during fine-tuning and are not trainable parameters.
> $\mathbf{L}$ and $\mathbf{R}$ define a global mapping from the low-rank latent space to the full parameter space, while only $\mathbf{A}^{t}$ is updated.
> We would like to clarify that this design is crucial for the communication efficiency.
> If we allow each client to update $\mathbf{L}$ and $\mathbf{R}$, and aggregate them on the server, then each client would need to upload and download all three matrices in each fine-tuning round.
> The per-round communication overhead (i.e., the number of transmitted model parameters) of each client would increase from $2rk$ to $2k(r+d_{\mathrm{out}}+d_{\mathrm{in}})$ with an extra $2k(d_{\mathrm{out}}+d_{\mathrm{in}})$.
> Since the LoRA rank $r$ is usually small, i.e., $r \ll d_{\mathrm{out}}$ and $r \ll d_{\mathrm{in}}$, aggregating all three matrices would increase the communication overhead by approximately $\frac{d_{\mathrm{out}}+ d_{\mathrm{in}}}{r}$ times, which is hundreds of times in our settings and would largely eliminate the communication savings of FLoRG.
>
> Regarding the sensitivity to initialization, we agree that the approaches to initializing L and R is crucial. Hence, we present an explicit study in Table 6. We compare our semi-orthogonal initialization with Kaiming and SVD initializations. The results show that the chosen semi-orthogonal scheme yields the best or comparable performance for both OPT-125M and RoBERTa-large, demonstrating the superiority of this semi-orthogonal initialization approach. More importantly, we would like to clarify that **under the chosen semi-orthogonal initialization scheme, our proposed approach can consistently achieve superior performance under randomly generated and fixed $\mathbf{L}$ and $\mathbf{R}$**. This can be seen from Table 6. In these experiments, the corresponding $\mathbf{L}$ and $\mathbf{R}$ are randomly and independently initialized across datasets, while our approach consistently outperforms the baselines.
>
> **Table 6:**
>
> | Base Model    | Initialization  | MRPC    | QQP      | MNLI      | QNLI      | WNLI     | RTE     |
> |--------------|------------------|---------|----------|-----------|-----------|----------|---------|
> | OPT-125M     | Semi-orthogonal  | **86.54** | 88.71   | **87.20** | **89.69** | **65.41** | 68.77  |
> | OPT-125M     | Kaiming          | 84.35   | **88.90** | 85.23    | 87.73    | 62.29    | **69.31** |
> | OPT-125M     | SVD              | 86.41   | 87.69   | 83.19    | 88.74    | 64.37    | 67.69  |
> | RoBERTa-large| Semi-orthogonal  | **89.87** | **91.27** | 91.39   | **92.48** | **66.41** | **71.40** |
> | RoBERTa-large| Kaiming          | 87.68   | 92.34   | 89.11    | 91.57    | 64.19    | 70.32  |
> | RoBERTa-large| SVD              | 88.70   | 90.37   | **91.49** | 91.45    | 65.08    | **71.40** |
>
> **Weakness 2:** Each round per layer requires eigendecomposition of Q and an SVD for Procrustes; scalability or latency with many layers or clients isn’t benchmarked.
>
> **Author's Response:** The server-side computation overhead consists of
> (i) performing an eigendecomposition of the $k\times k$ Gram matrix $\mathbf{Q}^{t+1}$ and (ii) solving the Procrustes alignment problem. In particular, the computation complexity of (i) is $\mathcal{O}(k^{3})$.
> The Procrustes alignment problem in (ii) has a closed-form solution as shown in Theorem 1: the optimal alignment matrix is $\mathbf{S}^{t,\star} = \mathbf{U}^{t+1}(\mathbf{V}^{t+1})^{\intercal}$, where matrices $\mathbf{U}^{t+1}\in\mathbb{R}^{r\times r^{\prime}}$, $\mathbf{V}^{t+1}\in\mathbb{R}^{r^{\prime}\times r^{\prime}}$, and $\boldsymbol{\Sigma}^{t+1}\in\mathbb{R}^{r^{\prime}\times r^{\prime}}$ are the SVD matrices of matrix $\mathbf{A}^{t}(\tilde{\mathbf{A}}^{t+1})^{\intercal}$.
> Therefore, the extra computation overhead is incurred by a single SVD in the low-rank subspace, with a computation complexity of $\mathcal{O}(krr^\prime + \min(r, r^\prime)^2\max(r,r^\prime))$.
> Since the LoRA rank $r$ is usually very small, i.e., $r\ll k$, the computation overhead of this additional SVD is much cheaper than that of the client-side forward pass/backpropagation and matrix decomposition, where these computations are also needed in the existing federated LoRA works (e.g., FeDeRA [1], FlexLoRA [2]).
> For a model with $L$ LoRA layers, the total per-round computation overhead of eigendecomposition and Procrustes alignment at the server is
> $\mathcal{O}(L k^3) + \mathcal{O}(L (krr^\prime + \min(r,r')^2 \max(r,r')))$. We include a brief discussion of the server-side computation overhead in Section 3.1 of the revised manuscript.

---

> ### Author Response · Authors · 2025-11-21
>
> **Author's Response Continued:** From this analysis, the per-round server computation overhead scales linearly with the number of LoRA layers and is independent of the number of clients, indicating that FLoRG is scalable even with many layers and many clients. From this complexity analysis, the per-round server computation overhead scales linearly with the number of LoRA layers and is independent of the number of clients, indicating that FLoRG is scalable even with many layers and many clients.
>
> **Question 1:** How robust is FLoRG if L,R are learned (slowly) or adapted per layer/round? Can you provide theory/ablation for updating L,R versus keeping them fixed?
>
> **Author's Response:** We thank the reviewer for the comment. Please refer to our response to weakness 1. The main ideas of the response are summarized as follows:
> - Updating both matrices $\mathbf{L}$ and $\mathbf{R}$ would increase the communication overhead significantly.
> - Our semi-orthogonal initialization scheme consistently matches or outperforms Kaiming and SVD initializations for both OPT-125M and RoBERTa-large. With randomly and independently generated semi-orthogonal $\mathbf{L}$ and $\mathbf{R}$ across datasets, our method still consistently outperforms all baselines, indicating it is robust rather than overly sensitive to a particular initialization.
>
> We would also like to clarify that, despite the semi-orthogonal initialization of the two matrices $\mathbf{L}$ and $\mathbf{R}$ being part of the contribution, the main contributions of this work are as follows:
> - **A new federated LoRA parameterization:** We introduce FLoRG, a single low-rank matrix parameterization with shared semi-orthogonal bases that fundamentally replaces the conventional two-matrix LoRA in FL. This design **eliminates aggregation error** among clients and **reduces the communication overhead significantly**.
> - **Procrustes alignment:** We identify **decomposition drift** as a critical issue in prior work and propose a Procrustes alignment algorithm that aligns the decomposed matrix between consecutive rounds, thereby stabilizing aggregation across clients.
> - **Convergence guarantee:** We provide a rigorous theoretical analysis which validates the effectiveness of Procrustes alignment on our proposed FLoRG. Our results show that the proposed alignment is not only empirically effective but also theoretically sound.
>
> **Question 2:** What are per-round costs of eigendecomposition + Procrustes across all LoRA layers at N>100 clients?
>
> **Author's Response:** We thank the reviewer for this question. The server-side cost has two components per round and per LoRA layer: (i) eigendecomposition of the $k \times k$ Gram matrix $\mathbf{Q}^{t+1}$, and (ii) Procrustes alignment. The eigendecomposition of a  $k \times k$ matrix has complexity $\mathcal{O}(k^3)$ per layer, independent of the number of clients $N$. For a model with $L$ LoRA layers, the total per-round server cost of eigendecompositions is therefore $\mathcal{O}(L k^3)$, matching the order of existing federated LoRA methods that already perform server-side matrix decompositions (e.g., FeDeRA [1], FlexLoRA [2]).
>
> The Procrustes alignment step introduces the only additional computation overhead to FLoRG. As shown in Theorem 1, the optimal alignment matrix is $\mathbf{S}^{t,\star} = \mathbf{U}^{t+1}(\mathbf{V}^{t+1})^{\intercal}$, where $\mathbf{U}^{t+1}\in\mathbb{R}^{r\times r^{\prime}}$, $\boldsymbol{\Sigma}^{t+1}\in\mathbb{R}^{r^{\prime}\times r^{\prime}}$, and $\mathbf{V}^{t+1}\in\mathbb{R}^{r^{\prime}\times r^{\prime}}$ come from the SVD of $\mathbf{A}^{t}(\tilde{\mathbf{A}}^{t+1})^{\intercal}$. Forming this product incurs a computation overhead of $\mathcal{O}(k r r^\prime)$, and the SVD on the resulting $r \times r^\prime$ matrix incurs a computation overhead of $\mathcal{O}(\min(r, r^\prime)^2 \max(r,r^\prime))$. Thus, the extra computation overhead of Procrustes alignment per round across all LoRA layers is $\mathcal{O}\!\big(L (k r r^\prime + \min(r,r')^2 \max(r,r'))\big)$, which is independent of $N$. Since the LoRA rank $r$ is very small in practice ($r \ll k$), this additional SVD is much cheaper than both the eigendecomposition overhead $\mathcal{O}(L k^3)$ and the client-side forward/backward computations, indicating that FLoRG remains scalable even with many layers and $N > 100$ clients.
>
> [1] Yuxuan Yan, Qianqian Yang, Shunpu Tang, and Zhiguo Shi. FeDeRA: Efficient fine-tuning of language models in federated learning leveraging weight decomposition. In _Proc. Advances Neural Info. Process. Syst. (NeurIPS)_, Vancouver, Canada, Dec. 2024.
>
> [2] Jiamu Bai, Daoyuan Chen, Bingchen Qian, Liuyi Yao, and Yaliang Li. Federated fine-tuning of large language models under heterogeneous tasks and client resources. In _Proc. Advances Neural Info. Process. Syst. (NeurIPS)_, Vancouver, Canada, Dec. 2024.
>
> We thank the reviewer for the comments. We hope that we have addressed the comments in a satisfactory manner.

---

### Official Review · Reviewer_F3oY · 2025-10-31

**Soundness:** 3
**Presentation:** 3
**Contribution:** 2
**Rating:** 4
**Confidence:** 5

**Summary:**

This paper proposes FLoRG, a framework for federated fine-tuning of LLMs using Low-rank Adaptation. The authors point out two primary challenges with existing federated LoRA methods:  Aggregation Error (caused by naively aggregating the LoRA matrices B and A separately), Decomposition Drift (caused by the fact that there is not unique decomposition matrix)
The authors tackle these challenges with a two-part solution:

* Gram Matrix Aggregation: Instead of LoRA's two matrices (B,A), FLoRG uses a single trainable low-rank matrix A and utilizes existing linear algebra techniques to convert the A matrix to the original dimension of the $\Delta W$.

* Procrustes Alignment: The server decomposes the aggregated weights, it performs a Procrustes alignment step. This solves an optimization problem to find an orthogonal matrix that best aligns the new A~t+1 with the matrix from the previous round At, thereby minimizing the "decomposition drift".

The paper provides a theoretical convergence analysis showing that the Procrustes alignment step results in a tighter convergence bound. The authors also empirically show that their method outperform four baselines on GLUE benchmarks.

**Strengths:**

* The paper is well-written. The authors did a good job categorising and explaining the existing problems.
* The algorithm performs better than the mentioned baselines.
* The authors provide a convergence analysis for FLoRG.

**Weaknesses:**

* Clarity on Communication Saving. I would appreciate it if the authors explained the communication saving part of their claim. Did they measure the communication compared to full matrix communication or other Federated LoRA methods?

* Server-Side Computational Overhead: The paper does not discuss the server-side computational cost, which appears to be substantial, especially doing matrix decomposition and solving optimization.

* The baselines are considerably basic. By just checking recent ACL and ICML conferences, I found recently accepted papers on Federated LoRA. The merits of the paper is not clear for me considering it is missing several works.

**Questions:**

* Information about the setting is missing. For example, what is the parameter for different levels of heterogeneity? How did you do LDA for datasets without labels?

* What is the federated learning setting, how many clients participate each round?

* Did you do hyperparameter search?

* Are the results averaged for different random seeds or they are done only for one seed?

Please also check the weakness section.

---

> ### Author Response · Authors · 2025-11-21
>
> **Weakness 1**: Clarity on Communication Saving. I would appreciate it if the authors explained the communication saving part of their claim. Did they measure the communication compared to full matrix communication or other Federated LoRA methods?
>
> **Author's Response**: We would like to clarify that in our experiments, communication overhead is measured as the total number of model parameters transmitted between all clients and the server over all fine-tuning rounds required to achieve the given target accuracies. The resulting end-to-end communication overhead for each method is summarized in Table 2 of the revised manuscript.
>
> **Table 2:**
> | Base Model     | Target Acc. | FLoRG            | FedIT           | FeDeRA          | FFA-LoRA        | FedSA-LoRA      | FedEx-LoRA        |
> |----------------|-------------|------------------|-----------------|-----------------|-----------------|-----------------|-------------------|
> | OPT-125M       | 80.00       | **8.2 × 10^6**   | 3.78 × 10^7     | 2.46 × 10^7     | 1.59 × 10^7     | 2.10 × 10^7     | 1.25 × 10^10      |
> | OPT-125M       | 85.00       | **1.07 × 10^7**  | -               | 4.2 × 10^7      | 2.75 × 10^7     | 3.61 × 10^7     | 1.77 × 10^10      |
> | RoBERTa-large  | 80.00       | **8.7 × 10^6**   | 4.68 × 10^7     | 3.02 × 10^7     | 1.85 × 10^7     | 2.59 × 10^7     | 2.08 × 10^10      |
> | RoBERTa-large  | 85.00       | **1.45 × 10^7**  | 8.12 × 10^7     | 5.17 × 10^7     | 3.35 × 10^7     | 4.42 × 10^7     | 2.96 × 10^10      |
>
> For the baseline schemes, the corresponding communication overhead is explained as follows:
> - **FedIT** and **FedEx-LoRA**: Each client exchanges the full LoRA module with the server, i.e., both low-rank matrices $\mathbf{B}$ and $\mathbf{A}$ in every round.
> - **FeDeRA**: Each client uploads and then receives the full LoRA module.
> - **FFA-LoRA**: Matrix $\mathbf{A}$ is frozen. Each client only exchanges $\mathbf{B}$ with the server.
> - **FedSA-LoRA**: Clients locally update both $\mathbf{B}$ and $\mathbf{A}$, but only matrix $\mathbf{A}$ is exchanged with the server.
> - **FedEx-LoRA**: Each client exchanges the full LoRA module, i.e., both low-rank matrices $\mathbf{B}$ and $\mathbf{A}$ in every round. In addition, each client receives an additional error-correction residual matrix in every round.
>
> In contrast, FLoRG uses a single low-rank matrix $\mathbf{A}$ on each client.
> Only this matrix is transmitted per round, while the corresponding Gram matrix and its decomposition are computed entirely on the server side without incurring extra communication overhead.
> As reported in Table 2, this design yields up to 2041$\times$ reduction in the total number of transmitted parameters compared to these federated LoRA baselines at the same target accuracies.
>
> **Weakness 2**: Server-Side Computational Overhead: The paper does not discuss the server-side computational cost, which appears to be substantial, especially doing matrix decomposition and solving optimization.
>
> **Author's Response**: We thank the reviewer for pointing this out.
> The server-side computation overhead consists of
> (i) performing an eigendecomposition of the $k\times k$ Gram matrix $\mathbf{Q}^{t+1}$ and (ii) solving the Procrustes alignment problem.
> In particular, the computation complexity of (i) is $\mathcal{O}(k^{3})$.
> The Procrustes alignment problem in (ii) has a closed-form solution as shown in Theorem 1: the optimal alignment matrix is $\mathbf{S}^{t,\star} = \mathbf{U}^{t+1}(\mathbf{V}^{t+1})^{\intercal}$, where matrices $\mathbf{U}^{t+1}\in\mathbb{R}^{r\times r^{\prime}}$, $\mathbf{V}^{t+1}\in\mathbb{R}^{r^{\prime}\times r^{\prime}}$, and $\boldsymbol{\Sigma}^{t+1}\in\mathbb{R}^{r^{\prime}\times r^{\prime}}$ are the SVD matrices of matrix $\mathbf{A}^{t}(\tilde{\mathbf{A}}^{t+1})^{\intercal}$.
> Therefore, the extra computation overhead is incurred by a single SVD in the low-rank subspace, with a computation complexity of $\mathcal{O}(krr^\prime + \min(r, r^\prime)^2\max(r,r^\prime))$.
> Since the LoRA rank $r$ is usually very small, i.e., $r\ll k$, the computation overhead of this additional SVD is much cheaper than that of the client-side forward pass/backpropagation and matrix decomposition, where these computations are also needed in the existing federated LoRA works (e.g., FeDeRA [1], FlexLoRA [2]).
> We have included a brief discussion of the server-side computation overhead in Section 3.1 of the revised manuscript.
>
> [1] Yuxuan Yan, Qianqian Yang, Shunpu Tang, and Zhiguo Shi. FeDeRA: Efficient fine-tuning of language models in federated learning leveraging weight decomposition. In _Proc. Advances Neural
> Info. Process. Syst. (NeurIPS)_, Vancouver, Canada, Dec. 2024.
>
> [2] Jiamu Bai, Daoyuan Chen, Bingchen Qian, Liuyi Yao, and Yaliang Li. Federated fine-tuning of
> large language models under heterogeneous tasks and client resources. In _Proc. Advances Neural
> Info. Process. Syst. (NeurIPS)_, Vancouver, Canada, Dec. 2024.

---

> ### Author Response · Authors · 2025-11-21
>
> **Weakness 3:** The baselines are considerably basic. By just checking recent ACL and ICML conferences, I found recently accepted papers on Federated LoRA. The merits of the paper is not clear for me considering it is missing several works.
>
> **Author's Response:** We thank the reviewer for the constructive comment. In Table 1, Figure 2, Table 2, Table 4, Table 5, and Table 7 of the revised manuscript, we have included results of FedEx-LoRA [1], which has been published in ACL 2025, as another baseline scheme. In FedEx-LoRA, each client transmits locally updated LoRA matrices $\mathbf{B}$ and $\mathbf{A}$ to the central server. The central server aggregates $\mathbf{B}$ and $\mathbf{A}$ separately and determines an additional residual matrix to eliminate the aggregation error.
>
> We would also like to emphasize that the existing baselines cover the main families of federated LoRA proposed in recent NeurIPS/ICLR/ACL works. These five baselines span the designs of **what** is transmitted (i.e. both low-rank matrices vs. one low-rank matrix) and **how** matrices are aggregated (i.e., direct averaging vs. matrix decomposition vs. exact aggregation). Our goal is to compare FLoRG against a representative and diverse set of federated LoRA baselines rather than exhaustively reimplement every recent variant.
>
> **Table 1:**
> | Base Model      | Dataset | FLoRG     | FedIT  | FeDeRA | FFA-LoRA | FedSA-LoRA | FedEx-LoRA |
> |-----------------|:--------|:---------:|:------:|:------:|:--------:|:----------:|:----------:|
> | OPT-125M        | MNLI    | **87.35** | 79.42  | 81.15  | 83.54    | 84.61      | 85.83      |
> | OPT-125M        | QNLI    | 89.52     | 84.18  | 86.71  | 87.93    | 88.69      | **89.88**  |
> | OPT-125M        | WNLI    | **65.28** | 58.45  | 59.34  | 62.61    | 62.83      | 64.15      |
> | OPT-125M        | RTE     | **68.92** | 61.08  | 64.51  | 66.02    | 67.39      | 68.27      |
> | RoBERTa-large   | MNLI    | **91.27** | 84.91  | 88.06  | 89.28    | 90.75      | 90.96      |
> | RoBERTa-large   | QNLI    | **92.58** | 87.49  | 90.07  | 90.96    | 91.54      | 92.13      |
> | RoBERTa-large   | WNLI    | **66.48** | 59.34  | 61.83  | 63.72    | 63.47      | 66.11      |
> | RoBERTa-large   | RTE     | **71.26** | 64.25  | 67.12  | 68.49    | 69.93      | 70.98      |
> | LLaMa-3.2-3B    | MNLI    | **93.15** | 87.24  | 89.83  | 91.05    | 92.38      | 92.74      |
> | LLaMa-3.2-3B    | QNLI    | 93.12     | 89.17  | 91.45  | 92.53    | **93.27**  | 93.05      |
> | LLaMa-3.2-3B    | WNLI    | **68.73** | 61.52  | 64.19  | 65.97    | 66.81      | 67.89      |
> | LLaMa-3.2-3B    | RTE     | **73.84** | 67.08  | 69.75  | 71.33    | 72.56      | 73.15      |
>
> **Table 2:**
> | Base Model     | Target Acc. | FLoRG            | FedIT           | FeDeRA          | FFA-LoRA        | FedSA-LoRA      | FedEx-LoRA        |
> |----------------|-------------|------------------|-----------------|-----------------|-----------------|-----------------|-------------------|
> | OPT-125M       | 80.00       | **8.2 × 10^6**   | 3.78 × 10^7     | 2.46 × 10^7     | 1.59 × 10^7     | 2.10 × 10^7     | 1.25 × 10^10      |
> | OPT-125M       | 85.00       | **1.07 × 10^7**  | -               | 4.2 × 10^7      | 2.75 × 10^7     | 3.61 × 10^7     | 1.77 × 10^10      |
> | RoBERTa-large  | 80.00       | **8.7 × 10^6**   | 4.68 × 10^7     | 3.02 × 10^7     | 1.85 × 10^7     | 2.59 × 10^7     | 2.08 × 10^10      |
> | RoBERTa-large  | 85.00       | **1.45 × 10^7**  | 8.12 × 10^7     | 5.17 × 10^7     | 3.35 × 10^7     | 4.42 × 10^7     | 2.96 × 10^10      |
>
> **Table 4:**
> | Rank  | Dataset | FLoRG     | FedIT  | FeDeRA | FFA-LoRA | FedSA-LoRA | FedEx-LoRA |
> |-------|:--------|:---------:|:------:|:------:|:--------:|:----------:|:----------:|
> | r = 2 | WNLI    | **60.55** | 55.57  | 56.30  | 57.50    | 59.14      | 58.32      |
> | r = 2 | RTE     | **65.82** | 58.41  | 61.19  | 62.88    | 64.30      | 62.79      |
> | r = 4 | WNLI    | **66.34** | 59.19  | 61.97  | 63.55    | 63.61      | 66.12      |
> | r = 4 | RTE     | **71.41** | 64.11  | 66.97  | 68.62    | 70.10      | 71.28      |
> | r = 8 | WNLI    | **68.83** | 61.70  | 63.52  | 65.10    | 66.47      | 68.02      |
> | r = 8 | RTE     | **72.10** | 64.78  | 66.99  | 68.02    | 70.61      | 71.30      |

---

> ### Author Response · Authors · 2025-11-21
>
> **Author's Response Continued:**
>
> **Table 5:**
> | Non-IIDness | Dataset | FLoRG     | FedIT  | FeDeRA | FFA-LoRA | FedSA-LoRA | FedEx-LoRA   |
> |-------------|:--------|:---------:|:------:|:------:|:--------:|:----------:|:------------:|
> | ρ = 0.1     | WNLI    | **60.12** | 53.07  | 54.21  | 56.14    | 57.74      | 58.84        |
> | ρ = 0.1     | RTE     | **65.30** | 55.60  | 59.19  | 61.20    | 60.75      | 61.67        |
> | ρ = 0.5     | WNLI    | **66.34** | 59.19  | 61.97  | 63.55    | 63.61      | 66.12        |
> | ρ = 0.5     | RTE     | **71.41** | 64.11  | 66.97  | 68.62    | 70.10      | 71.28        |
> | ρ = 1       | WNLI    | 67.83     | 61.70  | 63.52  | 64.33    | 65.61      | **68.31**    |
> | ρ = 1       | RTE     | 72.21     | 66.90  | 68.71  | 70.40    | 71.78      | **72.55**    |
>
> **Table 7:**
> | Client participation ratio | Dataset | FLoRG  | FedIT | FeDeRA | FFA-LoRA | FedSA-LoRA | FedEx-LoRA |
> |---------------------------|:--------|:------:|:-----:|:------:|:--------:|:----------:|:----------:|
> | 0.2                        | WNLI    | **58.42** | 54.15 | 55.20 | 56.30   | 57.10     | 56.46      |
> | 0.2                        | RTE     | **63.25** | 57.80 | 59.45 | 60.92   | 61.85     | 61.35      |
> | 0.5                        | WNLI    | **64.50** | 58.10 | 60.35 | 61.80   | 62.40     | 63.58      |
> | 0.5                        | RTE     | **69.15** | 62.50 | 65.20 | 66.75   | 68.20     | 68.61      |
> | 1                          | WNLI    | **66.34** | 59.19 | 61.97 | 63.55   | 63.61     | 66.12      |
> | 1                          | RTE     | **71.41** | 64.11 | 66.97 | 68.62   | 70.10     | 71.28      |
>
> [1] Raghav Singhal, Kaustubh Ponkshe, and Praneeth Vepakomma. FedEx-LoRA: Exact aggregation
> for federated and efficient fine-tuning of foundation models. In _Proc. Annu. Meeting Assoc. Com-
> put. Linguistics, Vienna, Austria, Jul. 2025.
>
> **Question 1:** Information about the setting is missing. For example, what is the parameter for different levels of heterogeneity? How did you do LDA for datasets without labels?
>
> **Author's Response:** We thank the reviewer for pointing this out.
> In our experiments, data heterogeneity across clients is modeled using a Dirichlet distribution over class labels, as described in Section 4.1.
> In particular, for each dataset, we sample client-specific label proportions $\boldsymbol{\pi}_n \sim \mathrm{Dir}(\rho)$, and allocate training samples to client $n$ according to $\boldsymbol{\pi}_n$.
> The scalar parameter $\rho>0$ controls the degree of data heterogeneity: a smaller $\rho$ yields more skewed label distributions, while a larger $\rho$ yields more balanced distributions.
> In the main context and Appendix A.7, we report results for $\rho\in\{0.1, 0.5, 1\}$ in Table 5 and Table 9, where the values of $\rho$ correspond to the cases from the most heterogeneous to mildly heterogeneous settings.
>
> Since all datasets in GLUE come with ground-truth labels, we do not perform Latent Dirichlet Allocation (LDA) on unlabeled data.
> In terms of the additional settings, we have included the hyperparameter settings in Appendix A.6 of the revised manuscript.
> For the question-answering task with SQuAD v1.1, we consider an iid setting.
>
> **Question 2:** What is the federated learning setting, how many clients participate each round?
>
> **Author's Response:** In our setting, we consider that all 20 clients participate in the fine-tuning.
> In the revised manuscript, we show the fine-tuning performance under varying client participation ratios.
> The results in Table 7 of the revised manuscript (please also refer to Table 7 in the response to Weakness 3) show that under different ratios of client participation, our proposed FLoRG outperforms the baseline schemes in most cases, which showcases its scalability.
>
> **Question 3:** Did you do hyperparameter search?
>
> **Author's Response:** Thank you for pointing this out. Yes, we did perform hyperparameter search. In Appendix A.6 of the revised manuscript, we have included the detailed settings of the hyperparameter search:
>
> In our experiments, the batch size is set to be 4 by searching over a range of {$2,4,8,16$}. Each local training epoch is set to be 1. We select the optimal learning rate from the range {$5e-4, 1e-4, 5e-5, 1e-5$}. The maximum sequence length is set to be 128, following the common practice in fine-tuning. We choose AdamW as the optimizer. We set the scaling factor $\alpha$ to be 16 for all algorithms.
>
> **Question 4:** Are the results averaged for different random seeds or they are done only for one seed?
>
> **Author's Response:** We thank the reviewer for the constructive comment.
> We conduct experiments with one seed. To address the reviewer's concern, in Table 1 of the revised manuscript (please also refer to Table 7 in the response to Weakness 3), we conduct experiments under 2 random seeds, which is the same as the settings in FlexLoRA [1]. And we report the average testing accuracy.

---

> ### Author Response · Authors · 2025-11-21
>
> **Author's Response Continued:** Results show that our proposed FLoRG still outperforms the baseline schemes under those four datasets in most cases.
>
> It is challenging to conduct experiments under multiple random seeds due to the limited rebuttal period.
> We will commit to including the experimental results under more random seeds upon acceptance of the paper.
>
> [1] Jiamu Bai, Daoyuan Chen, Bingchen Qian, Liuyi Yao, and Yaliang Li. Federated fine-tuning of
> large language models under heterogeneous tasks and client resources. In _Proc. Advances Neural
> Info. Process. Syst. (NeurIPS)_, Vancouver, Canada, Dec. 2024.
>
> We thank the reviewer for his/her constructive comments.
> We appreciate his/her time for reviewing the paper.
> We hope that we have addressed the comments in a satisfactory manner.

---

### Official Review · Reviewer_srrB · 2025-11-02

**Soundness:** 3
**Presentation:** 3
**Contribution:** 3
**Rating:** 6
**Confidence:** 3

**Summary:**

This paper addresses the issue that the federated aggregation of LoRA updates may not accurately reflect the intended global aggregation result. To this end, the authors propose FLoRG, which employs a single low-rank matrix for fine-tuning and aggregates its Gram matrix. Extensive experiments demonstrate its superiority over the existing works.

**Strengths:**

1. The paper presents a well-structured theoretical analysis with formal proofs, offering strong theoretical soundness and clear convergence guarantees.
2. It explores an interesting and under-studied problem—eliminating aggregation bias and decomposition drift in federated LoRA fine-tuning—introducing new insights into parameter-efficient federated learning.
3. The paper is clearly written, correctly annotated, and provides a thorough description of the proposed FLoRG framework, making it easy to follow and reproducible.

**Weaknesses:**

1. The paper does not address the partial client participation scenario, which is common in practical federated learning settings. Evaluating FLoRG under varying client availability would strengthen its applicability.
2. The experiments are conducted only on OPT-125M and RoBERTa-large, which are relatively dated compared to current state-of-the-art LLMs such as LLaMA-3 and Qwen-2.5. Using more recent backbones would better demonstrate the scalability and relevance of FLoRG.
3. The paper reports final accuracy and communication cost but does not include convergence curves showing performance versus communication rounds. Such a figure would provide clearer insights into the training dynamics and stability of FLoRG compared with baselines.

**Questions:**

**See weaknesses.**

---

> ### Author Response · Authors · 2025-11-21
>
> **Weakness 1**: The paper does not address the partial client participation scenario, which is common in practical federated learning settings. Evaluating FLoRG under varying client availability would strengthen its applicability.
>
> **Author's Response**: Thank you for pointing this out.
> In Table 7 of the revised manuscript, we compare the fine-tuning performance under varying client participation ratios.
> The results show that under different ratios of client participation, our proposed FLoRG outperforms the baseline schemes in most cases, demonstrating its scalability.
>
> **Table 7:**
> | Client participation ratio | Dataset | FLoRG  | FedIT | FeDeRA | FFA-LoRA | FedSA-LoRA | FedEx-LoRA |
> |:---------------------------:|:--------|:------:|:-----:|:------:|:--------:|:----------:|:----------:|
> | 0.2                        | WNLI    | **58.42** | 54.15 | 55.20 | 56.30   | 57.10     | 56.46      |
> | 0.2                        | RTE     | **63.25** | 57.80 | 59.45 | 60.92   | 61.85     | 61.35      |
> | 0.5                        | WNLI    | **64.50** | 58.10 | 60.35 | 61.80   | 62.40     | 63.58      |
> | 0.5                        | RTE     | **69.15** | 62.50 | 65.20 | 66.75   | 68.20     | 68.61      |
> | 1                          | WNLI    | **66.34** | 59.19 | 61.97 | 63.55   | 63.61     | 66.12      |
> | 1                          | RTE     | **71.41** | 64.11 | 66.97 | 68.62   | 70.10     | 71.28      |
>
> **Weakness 2**: The experiments are conducted only on OPT-125M and RoBERTa-large, which are relatively dated compared to current state-of-the-art LLMs such as LLaMA-3 and Qwen-2.5. Using more recent backbones would better demonstrate the scalability and relevance of FLoRG.
>
> **Author's Response**: We thank the reviewer for the constructive comment.
> In Table 1 of the revised manuscript, we have included Llama-3.2-3B for our experiments.
> The results show that our proposed FLoRG outperforms the baseline schemes
> under those four datasets in most cases.
>
> **Table 1:**
> | Base Model      | Dataset | FLoRG     | FedIT  | FeDeRA | FFA-LoRA | FedSA-LoRA | FedEx-LoRA |
> |-----------------|:--------|:---------:|:------:|:------:|:--------:|:----------:|:----------:|
> | OPT-125M        | MNLI    | **87.35** | 79.42  | 81.15  | 83.54    | 84.61      | 85.83      |
> | OPT-125M        | QNLI    | 89.52     | 84.18  | 86.71  | 87.93    | 88.69      | **89.88**  |
> | OPT-125M        | WNLI    | **65.28** | 58.45  | 59.34  | 62.61    | 62.83      | 64.15      |
> | OPT-125M        | RTE     | **68.92** | 61.08  | 64.51  | 66.02    | 67.39      | 68.27      |
> | RoBERTa-large   | MNLI    | **91.27** | 84.91  | 88.06  | 89.28    | 90.75      | 90.96      |
> | RoBERTa-large   | QNLI    | **92.58** | 87.49  | 90.07  | 90.96    | 91.54      | 92.13      |
> | RoBERTa-large   | WNLI    | **66.48** | 59.34  | 61.83  | 63.72    | 63.47      | 66.11      |
> | RoBERTa-large   | RTE     | **71.26** | 64.25  | 67.12  | 68.49    | 69.93      | 70.98      |
> | LLaMa-3.2-3B    | MNLI    | **93.15** | 87.24  | 89.83  | 91.05    | 92.38      | 92.74      |
> | LLaMa-3.2-3B    | QNLI    | 93.12     | 89.17  | 91.45  | 92.53    | **93.27**  | 93.05      |
> | LLaMa-3.2-3B    | WNLI    | **68.73** | 61.52  | 64.19  | 65.97    | 66.81      | 67.89      |
> | LLaMa-3.2-3B    | RTE     | **73.84** | 67.08  | 69.75  | 71.33    | 72.56      | 73.15      |
>
> **Weakness 3**: The paper reports final accuracy and communication cost but does not include convergence curves showing performance versus communication rounds. Such a figure would provide clearer insights into the training dynamics and stability of FLoRG compared with baselines.
>
> **Author's Response**: We thank the reviewer for the constructive comment. we present the convergence
> curves of the training loss and testing accuracy in Fig. 2 of the revised manuscript.
>
> We thank the reviewer for his/her constructive comments.
> We appreciate his/her time for reviewing the paper.
> We hope that we have addressed the comments in a satisfactory manner.

---

> > ### Comment · Reviewer_srrB · 2025-11-25
> >
> > Thank you for your response, and I appreciate your efforts in rebuttal. I find it helpful and decide to increase my confidence to support that the paper could be weakly accepted.

---

> > > ### Author Response · Authors · 2025-11-26
> > >
> > > Thank you very much for your follow-up and for the thoughtful review throughout the process. We are glad that our clarifications helped address your concerns, and we truly appreciate your increased confidence and support for the paper.

---

### Author Response · Authors · 2025-12-01

We would like to thank the area chair and all reviewers for their careful reading and feedback. We have revised the manuscript to address reviewers' main concerns. We summarize the key issues each reviewer raised and the corresponding changes in the revised manuscript as follows:

**To Reviewer 1:**
- **Partial participation scenario:** We compare the performance under varying client participation ratios in Table 7 to demonstrate the robustness of FLoRG to client availability.
- **Larger base model:** We additionally evaluate FLoRG on LLaMa-3.2-3B. The results are shown in Table 1 to demonstrate the scalability of FLoRG.
- **Visualization of loss and accuracy:** We included the convergence curve of both loss and accuracy in Figure 2.


**To Reviewer 2:**
- **Detailed comparison of communication cost, computation overhead, and memory usage:** In Table 2, we compare the total number of transmitted parameters to achieve the target accuracy. In Table 8, we report the per-round FLOPs for clients and server and the memory usage of each baseline. Results highlight the efficiency of FLoRG.
- **More recent baseline:** We added **FedEx-LoRA** (ACL 2025) as an additional baseline in all main and ablation experiments. The results are shown in Tables 1, 2, 4, 5, 7, and 8, as well as Figure 2.
- **More datasets and experiment details:** We clarify the setting of data heterogeneity. We show the batch size, learning rate search range, maximum sequence length, number of local epochs, and optimizer in Appendix A.6. We also report the use of two random seeds and their corresponding average performance.

**To Reviewer 3:**
- **Robustness of initialization of $\mathbf{L}$ and $\mathbf{R}$:** We emphasize that matrices $\mathbf{L}$ and $\mathbf{R}$ are shared semi-orthogonal bases that make FLoRG compatible with arbitrary parameter shapes while keeping the global Gram-matrix aggregation simple. Updating them will incur significant communication overhead. Experimental results show that the proposed approach consistently achieves superior performance under randomly generated $\mathbf{L}$ and $\mathbf{R}$.
- **Computation complexity of eigendecomposition and SVD for Procrustes alignment and scalability issue of the number of clients $N$:** We present the computation complexity of both matrix decompositions, which shows that this additional computation complexity is only incurred by SVD for Procrustes alignment and is independent from the number of clients $N$.

**To Reviewer 4:**
- **Larger base model and additional task:** Same as Reviewer 1, we additionally evaluate FLoRG on LLaMa-3.2-3B. The results are shown in Table 1 to demonstrate the scalability of FLoRG.
In addition, we use SQuAD v1.1 as the question-answering dataset. The results are shown in Table 9 to demonstrate the superiority of FLoRG.
- **Expressiveness of only updating $\mathbf{A}$:** We clarified that in FLoRG, each layer’s update is parameterized by a single $\mathbf{A}$, which is mapped through shared semi-orthogonal bases $\mathbf{L}$ and $\mathbf{R}$. This preserves a rich low-dimensional subspace in the original parameter space while (i) reducing the number of trainable and transmitted parameters and (ii) avoiding scale imbalance and optimizer-sensitivity issues known for the two-matrix $\mathbf{BA}$ parameterization. Results show that FLoRG matches or outperforms baselines that update two low-rank matrices, providing evidence that updating only $\mathbf{A}$ is sufficient in practice.
- **Detailed comparison of communication cost, computation overhead, and memory usage:** Same as Reviewer 2, in Table 2, we compare the total number of transmitted parameters to achieve the target accuracy. In Table 8, we report the per-round FLOPs for clients and server and the memory usage of each baseline. Results highlight the efficiency of FLoRG.
- **Centralized v.s. federated setting:** We clarified that if we run FLoRG on a single machine, the local update reduces to learning a single matrix, which is closely related to centralized single-matrix adapters such as SingLoRA. FLoRG handles rectangular parameter matrices via shared $\mathbf{L}$ and $\mathbf{R}$. We highlight that the main advantages of FLoRG appear in the FL setting, including communication savings, an aggregation mechanism which eliminates the aggregation error and mitigates the decomposition drift known in current federated LoRA works, and the avoidance of the $\mathbf{BA}$ scale imbalance and optimizer issues.
- **Dataset usage and partitioning:** We clarify the datasets and the Dirichlet distribution for non-iid dataset partitioning in Section 4.1.

Finally, we thank the area chair for carefully reading this response and the reviewers for their constructive comments. We hope that we have addressed their comments in a satisfactory manner.

---

### Meta-Review · Area_Chair_zuCD · 2026-01-07

**Summary:**

The reviewers' concerns are centered around the evaluation of the proposed method with larger models and more complex tasks, and additional scenarios(e.g. Partial participation), comparing it with additional baselines, clarification on computation and communication costs and experimental details.

**Reviewer Concerns:**

Across all reviewers' comments, I find the rebuttal has addressed the concerns very well. What is outstanding is additional experimental results on more complex tasks and enhancing the statistical significance of results by conducting experiments with more runs. I hope the authors will continue to revise the paper by adding these results in their final draft.

**Reviewer Scores:**

Reviewer srrB keeps a positive score and increase his/her confidence to support that the paper could be weakly accepted.
Reviewer F3oY's concerns appear to be addressed adequately so I think this reviewer might raise his/her score to a positive value.
Reviewer fXbx's concerns appear to be addressed adequately so I think this reviewer might keep or raise his/her score.
Reviewer kKoy's concerns on evaluation on more tasks is addressed with  additional experiments on one task. The other concerns appear to be addressed adequately so I think this reviewer might keep or raise his/her score.

---

### Decision · Program_Chairs · 2026-01-26

Accept (Poster)